# Role of CB1 Cannabinoid Receptors in Vascular Responses and Vascular Remodeling of the Aorta in Female Mice

**DOI:** 10.3390/ijms242216429

**Published:** 2023-11-17

**Authors:** Bálint Bányai, Zsolt Vass, Stella Kiss, Anikó Balogh, Dóra Brandhuber, Gellért Karvaly, Krisztián Kovács, György L. Nádasy, László Hunyady, Gabriella Dörnyei, Eszter Mária Horváth, Mária Szekeres

**Affiliations:** 1Department of Physiology, Faculty of Medicine, Semmelweis University, 37-47 Tűzoltó Street, 1094 Budapest, Hungary; banyai.balint@phd.semmelweis.hu (B.B.); k.stella1114@gmail.com (S.K.); nadasy.gyorgy@med.semmelweis-univ.hu (G.L.N.); hunyady.laszlo@med.semmelweis-univ.hu (L.H.); horvath.eszter.maria@semmelweis.hu (E.M.H.); 2Department of Morphology and Physiology, Faculty of Health Sciences, Semmelweis University, 17 Vas Street, 1088 Budapest, Hungary; vass.zsolt@semmelweis.hu (Z.V.); anikobalogh0@gmail.com (A.B.); bdora995@gmail.com (D.B.); 3Department of Laboratory Medicine, Faculty of Medicine, Semmelweis University, 4 Nagyvárad Square, 1089 Budapest, Hungary; karvaly.gellert_balazs@med.semmelweis-univ.hu (G.K.); kovacs.krisztian1@med.semmelweis-univ.hu (K.K.); 4Institute of Enzymology, HUN-REN Research Centre for Natural Sciences, 2 Magyar Tudósok Körútja, 1117 Budapest, Hungary

**Keywords:** endocannabinoid, cannabinoid type 1 receptor, estrogen receptor, vascular remodeling, endothelium

## Abstract

Both the endocannabinoid system (ECS) and estrogens have significant roles in cardiovascular control processes. Cannabinoid type 1 receptors (CB_1_Rs) mediate acute vasodilator and hypotensive effects, although their role in cardiovascular pathological conditions is still controversial. Estrogens exert cardiovascular protection in females. We aimed to study the impact of ECS on vascular functions. Experiments were performed on CB_1_R knockout (CB_1_R KO) and wild-type (WT) female mice. Plasma estrogen metabolite levels were determined. Abdominal aortas were isolated for myography and histology. Vascular effects of phenylephrine (Phe), angiotensin II, acetylcholine (Ach) and estradiol (E2) were obtained and repeated with inhibitors of nitric oxide synthase (NOS, Nω-nitro-L-arginine) and of cyclooxygenase (COX, indomethacin). Histological stainings (hematoxylin-eosin, resorcin-fuchsin) and immunostainings for endothelial NOS (eNOS), COX-2, estrogen receptors (ER-α, ER-β) were performed. Conjugated E2 levels were higher in CB_1_R KO compared to WT mice. Vasorelaxation responses to Ach and E2 were increased in CB_1_R KO mice, attenuated by NOS-inhibition. COX-inhibition decreased Phe-contractions, while it increased Ach-relaxation in the WT group but not in the CB_1_R KO. Effects of indomethacin on E2-relaxation in CB_1_R KO became opposite to that observed in WT. Histology revealed lower intima/media thickness and COX-2 density, higher eNOS and lower ER-β density in CB_1_R KO than in WT mice. CB_1_R KO female mice are characterized by increased vasorelaxation associated with increased utilization of endothelial NO and a decreased impact of constrictor prostanoids. Our results indicate that the absence or inhibition of CB_1_Rs may have beneficial vascular effects.

## 1. Introduction

Endogenously produced cannabinoids—endocannabinoids (eCBs)—take part in various physiological functions. They serve as endogenous ligands for cannabinoid receptors and participate in tissue-specific regulatory mechanisms discovered first in the nervous system. Their receptors include type 1 cannabinoid receptors (CB_1_Rs), which are characteristically present in neural tissues, and type 2 cannabinoid receptors (CB_2_Rs), which occur mostly in immune cells [1]. In addition to their key functions in the central neural system, the role of cannabinoid receptors in the peripheral tissues is also increasingly a focus of interest [1,2,3,4,5,6,7]. Cannabinoid receptors were originally named after their affinity for Δ9-tetrahydrocannabinol (THC), the main active ingredient of the extracts of *Cannabis sativa*. In addition to CB_1_Rs and CB_2_Rs, other receptors have also been found to respond to cannabinoids, such as GPR55 and Transient Receptor Potential cation channel Vanilloid-1 (TRPV-1) [3,8,9,10]. Of the two cannabinoid receptor subtypes, cannabinoid receptor CB_1_Rs are characteristically present in the central neural system, typically in the presynaptic locations modulating synaptic transmission [1,4]. During stimulation by neurotransmitters acting on G protein-coupled receptors (GPCRs) such as glutamate and acetylcholine, endocannabinoid-mediated CB_1_R activation exerts depolarization-induced retrograde synaptic inhibition [2,6,11]. To date, several endocannabinoid compounds have been identified, including arachidonoyl ethanolamide (anandamide, AEA) and 2-arachidonoylglycerol (2-AG) [1,4,5].

The endocannabinoid system (ECS) involves endocannabinoids, their receptors and enzymes that modulate their production (e.g., diacylglycerol (DAG) lipase, NAPE-PLD) and degradation (e.g., monoacylglycerol (MAG) lipase, fatty acid amide hydrolase (FAAH), etc.). Production of the endocannabinoid 2-AG is catalyzed by diacylglycerol DAG lipase (DAGL), and its degradation is due to monoacylglycerol MAG lipase (MAGL). Degradation of AEA is catalyzed by FAAH [10,12].

It has been shown that the cannabinoid system contributes to cardiovascular, inflammatory, gastrointestinal and metabolic regulatory mechanisms [2,3,4,5,9,13]. In the cardiovascular system, negative inotropic, vasodilator and hypotensive actions of cannabinoids have been reported [3]. It was found earlier in our laboratory that calcium signal-generating hormones such as angiotensin II (Ang II) induced CB_1_R coactivation, which effect was inhibited by DAG-lipase inhibitors. It has been suggested that DAG, which is generated from phosphoinositides, is converted to 2-AG by DAGL during the signaling of Ca^2+^-mobilizing hormones and neurotransmitters [6,7]. Functional studies have revealed the existence of this mechanism in vascular tissues and demonstrated the modulatory effects of eCBs in GPCR signaling-induced vasoconstriction [2,5,14,15,16]. Also, in vascular tissue, the release of endocannabinoids has been directly detected [5,17]. Endocannabinoids are able to influence vascular remodeling processes [18]. There is an increasing hope that the growing number of studies with compounds modulating the ECS may lead to novel therapeutic approaches in a number of metabolic and cardiovascular disorders [4,9,11,13].

Significant gender differences have been described in several vascular functions, among others in the aorta, in coronary arteries and skeletal muscle arterioles [19,20,21]. Vascular functions are substantially modified by endothelial factors such as nitric oxide (NO) and by prostanoids (PGs), where the latter can be both vasodilatory and constrictor PGs [16,22,23,24,25]. There are marked gender differences in vascular contractility and its endothelial modulation. For example, in female animals, contractility of coronary arteries was decreased, while estrogen replacement therapy after ovariectomy augmented NO-mediated dilation [19,20]. Estrogen induces acute nongenomic vasodilatory effects, which were shown to be mostly mediated by endothelial NO [26,27,28]. In female vessels endothelium-dependent vasodilatory functions may be augmented by estrogens, inducing vascular protective effects before menopause and applied in the frame of estrogen replacement therapy after menopause [20,29,30]. Sex differences can exist in the cannabinoid sensitivity of blood vessels: lifetime cannabis use induced increased vascular stiffness in males [31].

Earlier investigations revealed that the ECS is also involved in the control processes of the female reproductive system. A delicate balance between endocannabinoid production and degradation, as well as a well-regulated CBR activity, are required for the optimal function of the reproductive tract and the hypothalamic–pituitary–ovarian (HPO) axis [32,33]. Among cannabis (marijuana) and synthetic cannabinoid users, sexual endocrine disturbance emerges frequently, causing cycle abnormalities and infertility [33,34]. THC, the main component of marijuana, blocks the release of gonadotropin releasing hormone (GnRH) from the hypothalamus and thus the production of luteinizing hormone (LH) by the adenohypophysis [33,34,35]. In women using marijuana, a disturbed menstrual cycle, reduced number of oocytes harvested during in vitro fertilization and higher risk of premature delivery, even abortions, have been described [36]. The ECS is likely to play a role in control of the organization of the endometrial cycle. Several components of the ECS, including CB_1_R, CB_2_R, the degrading enzyme FAAH, have been found to be highly expressed in the female reproductive tract, their function has been only partially explored [37].

There is an interplay between the endocannabinoid system and the female reproductive hormones. 7β-estradiol (E2) can modulate CB_1_R expression and the degradation processes of the endocannabinoid anandamide [38]. The endocannabinoid system is involved in the control of ovarian follicle maturation [39].

Considering the vascular effects of the two estradiol receptor (ER) subtypes, it has been shown earlier that the estrogen receptor alpha (ER-α) conveys the important vasorelaxation effects of this hormone [40]. In recent years, however, the functions of the estrogen receptor beta (ER-β) in the vascular tissue have also been demonstrated: ER-β activation has both direct vascular effects and can act indirectly through energy homeostasis [41] as well as through other components of the cardiovascular system [42,43,44].

Our aim was to study the impact of ECS and CB_1_ receptor activation on vascular functions. Aortic tissue of CB_1_R knockout (CB_1_R KO) mice was used to demonstrate the impact of missing receptor function. KO animals are the proper tools to reveal the full functional significance of a receptor function throughout the lifecycle. We have a good reason to think that several effects of missing function can be successfully reproduced pharmacologically in the presence of intact receptor molecules by administration of inhibitors. Changes in contractility and endothelial modulation of contractility through NO and vasoactive prostanoids were examined with specific blockers. With histology examinations, we aimed to reveal the potential role of CB_1_Rs in structural vascular remodeling processes, the impact of missing receptors on such important parameters as intima-media thickness ratio, elastic tissue and contractile protein densities, the expression of endothelial nitric oxide synthase (eNOS, which synthesizes NO) and cyclooxygenase (COX, which synthesizes prostanoids), estrogen and thromboxane receptor expression levels in the aortic tissue of female mice. Special attention has been given to a potential interplay between the ECS and estrogens. To investigate the possible involvement of ECS in the function of the HPO axis, we also determined plasma estrogen and metabolite levels.

## 2. Results

### 2.1. Effects of Presence of CB1 Receptors on Vascular Contraction and Relaxation Responses

Concentration–response curves in response to two vasoconstrictors, the alpha adrenergic receptor agonist phenylephrine (Phe) and the angiotensin type 1 (AT_1_) receptor agonist Ang II, were obtained on abdominal aortic segments of female wild-type (WT) and CB_1_R KO mice (in concentrations of 10^−9^–10^−5^ mol/L of Phe and 10^−9^–10^−7^ mol/L of Ang II). Both Phe and Ang II induced dose-dependent vasoconstrictions. Vasoconstrictor responses did not show significant differences between the two genetic groups (Figure 1A,B). Despite similar vasoconstrictions, responses to vasodilators, acetylcholine (Ach) and estradiol were significantly different in the two genetically different strains. In both cases, higher dilations were obtained in the aortas of CB_1_R KO mice compared to the WT group (Ach: 10^−8^ and 10^−7^ mol/L, *p* = 0.002 and 0.044, respectively, E2: 10^−5^ mol/L, *p* = 0.008, 2-way ANOVA with Bonferroni post hoc test, Figure 1C,D). The CB_1_R agonist WIN 55,212-2 induced significant vasodilation in the aortas of WT female mice (12.1 ± 3.4%, *n* = 5), which was not observed in the CB_1_R KO mice (2.3 ± 3.3%, *n* = 6). This difference was statistically significant (*p* = 0.033, in unpaired *t*-test, Figure 1E).

### 2.2. Effects of Specific Inhibitors on Contraction and Relaxation Vascular Responses in WT (CB_1_R+/+) and in CB_1_R KO (CB_1_R−/−) Female Mice

To reveal potential differences in vascular tone control mechanisms, contractions to phenylephrine, relaxations in response to Ach and estradiol in the presence of the inhibitor of NO synthesis Nω-nitro-L-arginine (LNA) and of the COX inhibitor indomethacin (INDO) in the two genetically different groups have been tested. Specific inhibitors INDO and LNA, as well as “vehicle” did not modify baseline vascular tone (vehicle: by −0.01 ± 0.26 mN, *n* = 9, INDO: by 0 ± 0.25 mN, *n* = 10, LNA: by 0.15 ± 0.37 mN, *n* = 10). Results are shown in the panels of Figure 2. While, in Figure 3, the contractions and relaxations achieved in the two genetically different groups are directly compared.

Contraction dose–response to Phe was not affected by the presence of LNA in the bathing solution in either genetic group (Figure 2A,B and Figure 3A). However, Phe contractions were attenuated in the presence of INDO in the bathing solution both in the WT (at 10^−6^ to 10^−5^ mol/L, *p* < 0.001, interaction: *p* = 0.006, 2-way ANOVA, Figure 2A) and in the CB_1_R KO (at 10^−6^ to 10^−5^ mol/L, *p* < 0.001, interaction: *p* < 0.001, 2-way ANOVA, Figure 2B) group. This contraction–attenuation was similar in the two strains (Figure 3B).

As could be expected, in WT mice the concentration-dependent Ach vasorelaxation was significantly attenuated by the nitric oxide synthase (NOS) inhibitor LNA (*p* < 0.05 at 10^−8^ mol/L, *p* < 0.001 at doses of 10^−7^–10^−6^ mol/L, interaction: *p* < 0.001, 2-way ANOVA). Ach relaxation was significantly enhanced by the COX inhibitor INDO (*p* < 0.05 at 10^−7^ mol/L and *p*= 0.026 at 10^−6^ mol/L, Figure 2C) in WT animals. Similar to the WT mice, in CB_1_R KO mice, Ach-induced relaxation was also significantly attenuated by LNA (*p* < 0.001 at doses of 10^−8^–10^−6^ mol/L, interaction: *p* < 0.001), but the effect of INDO here could not be confirmed (Figure 2D). Direct comparison of the two strains demonstrated no difference in LNA effect (Figure 3C), while the effect of INDO on Ach relaxation was statistically different (*p* = 0.026 with two-way ANOVA, Bonferroni post hoc test, *p* < 0.05 at 10^−7^ and 10^−6^ mol/L Ach, Figure 3D). Similarly to the effect of INDO in Ach-induced relaxation response, we have found that the specific thromboxane-prostanoid (TP) receptor (TP-R) blocker SQ 29,548 augmented Ach-induced relaxation in WT, but not in KO mice (Appendix A).

Both in WT and CB_1_R KO mice, LNA effectively reduced E2-relaxation (*p* < 0.001 at 10^−5^ mol/L in both groups, Figure 2E,F), while INDO in WT slightly elevated, and in CB_1_R KO animals slightly reduced E2 relaxation effect, but this did not reach the level of statistical significance (Figure 2E,F). Slight differences in E2 relaxation in the two genetic groups with INDO (Figure 3F) were statistically significant if normalized to the original relaxation effect of E2 (Figure 3G).

Calculated pharmacological parameters of Emax, EC50 and pEC50 are indicated in Appendix A.

### 2.3. Estrogen Metabolite Levels

Free and conjugated estrone, 2-hydroxyestrone and conjugated 4-hydroxyestrone did not reach threshold levels of determination in either of the samples. Free estradiol (fE2) levels did not show a significant difference between WT (*n* = 24) and CB_1_R KO (*n* = 17) female mice. However, conjugated estradiol (cE2) levels were significantly higher (*p*= 0.039) in the CB_1_R KO (*n* = 11) female mice than their WT counterparts (*n* = 16). Although free 4-hydroxyestrone (f4OH-E1) levels reached higher values in CB_1_R KO mice (*n* = 13) compared to WT female mice (*n* = 14), it did not reach the level of statistical significance (Figure 4).

### 2.4. Histology and Immuno-Histochemistry

The intima–media ratio was significantly lower in the CB_1_R KO group (*n* = 10) compared to the control group (*n* = 6) due to the thickening of the medial layer. The overall thickness of the wall was also increased (Figure 5A–C). There was no significant difference in elastic density as studied by resorcin fuchsin (RF) staining between the two strains (Figure 5D,E). α-smooth muscle actin (α-SMA) stainings also did not show a difference between the CB_1_R KO and the WT group (Figure 5F,G).

### 2.5. Vasoactive Markers/Molecular Contributors of Vascular Functions

There was a significantly increased eNOS protein density in the wall of the aortas of the CB_1_R KO group (*n* = 14) compared to the control group (*n* = 9) (Figure 6A,B). In the case of the cyclooxygenase type 2 (COX-2) in the endothelium, an opposite change was observed (*n* = 8 and 12 for the control and CB_1_R KO groups, respectively) (Figure 6C,D). No difference in the expression of the TP receptor protein density was observed (*n*= 6 and 13 for the control and CB_1_R KO groups, respectively) (Figure 6E,F). Although TP-R density was similar between groups, there is a tendency to higher levels of thromboxane synthase in WT compared to CB_1_R KO group (Appendix A).

### 2.6. Estrogen Receptor Expression

We examined the density of the estrogen receptor alpha (ER-α) in both the endothelial and medial layers of the aortic wall and found no significant difference (Figure 7A–C). In the case of the estrogen receptor beta (ER-β), however, we found a decreased expression in both the endothelial and medial layers in CB_1_R KO mice (Figure 7D–F).

### 2.7. Nitrative Stress

There was no 3-Nitrotyrosine density difference in the aortic walls between the control and the CB_1_R KO group (Figure 8A,B).

## 3. Discussion

### 3.1. Main Functional Observations and Their Molecular Background

To reveal the effects of eCBs and their CB_1_R receptors, in our study, we have compared functional and structural vascular remodeling in the aortas of wild-type (CB_1_R+/+) and CB_1_R KO (CB_1_R−/−) female mice. The main findings of our study are as follows. Although contraction responses were not different in the two groups, Ach- and estradiol-induced vasorelaxations were significantly higher in CB_1_R KO mice compared to WT. *This proves the functional involvement of endocannabinoids in both relaxation processes.* Ach-induced relaxation was effectively inhibited by the NOS blocker LNA in both strains. On the other hand, Ach relaxation response was augmented by the COX inhibitor INDO in the WT mice, but not in CB_1_R KO animals. Similar results have been found with the specific TP-R blocker SQ 29,548. These results indicate that Ach-induced relaxation is mediated through nitric oxide in both groups (as it could be expected) and that *constrictor prostanoids contribute to the vascular effects in WT but not in CB_1_R KO mice.* We also found that conjugated estradiol levels were higher and f4OH-E1 levels had a tendency to increase in CB_1_R KO compared to WT mice, indicating that *CB_1_ receptors have a role in controlling serum estrogen levels.* Effects of NOS inhibition significantly reduced estradiol induced vasodilation in both groups. During COX inhibition, estradiol relaxation was slightly augmented in WT, while in KO mice it was slightly depressed. Comparison of normalized values revealed statistical difference between the two groups, estradiol relaxation of CB_1_R KO animals being more reduced. *This proves the functional contribution of both endocannabinoids and endogenous prostanoids to estradiol induced vascular relaxation.*

To test the functional responses of CB_1_ receptors, we found that WIN 55,212-2, a CB_1_ receptor agonist, induced significant vasodilation in CB_1_R wild-type mice, which was not observed in CB_1_R knockout mice.

Our histology results indicate significant vascular remodeling in CB_1_R KO mice compared to WT. The intima–media ratio of the CB_1_R KO group was significantly lower compared to the control group as a result of increased media thickness (with unaltered intima thickness) in the absence of CB_1_ receptors. We can make the pathologically very important conclusion that *CB_1_Rs and endocannabinoids are important contributors to morphological remodeling of the vascular (aortic) wall.* While the geometry of the wall did change, it seems that control processes determining the histological composition, the expression of the connective tissue component elastin and of the contractile protein alfa-smooth muscle actin, were left unaltered. No significant difference was found between the two strains in TP-R receptor and ER alpha receptor staining. *Differences in endogenous prostanoid effects we observed thus are not connected with altered receptor expression.* However, in the CB_1_R KO group, significant reduction was found in ER beta receptor expression. This may be related to the elevated estrogen levels detected in the receptor deficient group, suggesting a potential negative feedback mechanism. In addition, *endocannabinoids and CB_1_Rs can be considered important components in vascular responses to sex hormones.*

Products of eNOS and COX-2 enzymes were behind the vascular functional (relaxation) changes described above. Performing immunohistochemistry, we found that eNOS density was significantly elevated in CB_1_R KO animals. This is in direct accord with our functional observations that Ach- and E2-induced relaxations were more powerful in the same strain of animal. We can conclude that *endocannabinoids acting through CB_1_Rs control NOS expression in the aortic wall, moderating the amount of available enzyme molecules and reducing the extent of endothelial dilation.* COX-2 density was significantly lower in the aortic wall when CB_1_ receptors were knocked out. This is again in good agreement with our functional observations, according to inhibition of COX augmented Ach relaxation in WT mice but not in CB_1_R KO mice. Similar difference could be proven for E2 dilation. That proves *the moderation of endothelial relaxation through endogenous vasoconstrictor prostanoids by endocannabinoids in the presence of CB_1_Rs.* We tested the density of the receptor of the most powerful vasoconstrictor prostanoid, thromboxane TP-R, but no differences between the two strains could be proven. However, thromboxane synthase activity had an increased tendency in WT compared to KO mice. There was no difference in the nitrative damage to the wall proteins either.

### 3.2. Vascular Effects of Cannabinoids and Endocannabinoid Signaling

In vascular tissue cannabinoid receptor-mediated signaling mechanisms may influence vascular tone in healthy or in pathological conditions [3,5,11,45]. Targets of cannabinoid agonists in the vascular tissue are endothelial and vascular smooth muscle cells and perivascular neurons [46,47,48,49]. Previous studies revealed that cannabinoids and synthetic agonists such as AEA, THC, WIN 55,212-2 induce vasodilation in different vascular beds, such as in aorta, coronary and cerebral arteries [5,14,46,47,48,49,50]. The extent of vasodilatory effect of cannabinoids may vary among vessel types; a higher relaxation effect can be obtained on resistance arteries [14,51] than on the aorta [5,46].

Endocannabinoids mediate acute vasodilatory and hypotensive effects upon CB_1_R stimulation, which effect is via G_i/o_ protein-coupled signaling [3,5,46,52,53]. Similarly to the mechanism of retrograde synaptic transmission, endocannabinoid release has been detected during calcium generating GPCR agonists that can modulate vasoconstriction by a negative feedback mechanism [2,15]. Such mechanisms have been observed in different vascular beds such as in pulmonary, cerebral and coronary arteries, gracilis arteriole or in the aorta [5,14,16,45,54], which suggested the existence of a continuous vasodilator tone in the vascular wall through endocannabinoid signaling. Ang II-induced vasoconstriction was augmented by inhibition CB_1_Rs and also of DAGL, while it was attenuated by inhibition of MAGL, suggesting that locally produced 2-AG activates vascular CB_1_Rs [5]. In spontaneously hypertensive rats, the inhibition of anandamide degrading enzyme fatty acid amide hydrolase by URB597 augmented AEA level and improved vascular endothelial functions in small mesenteric arteries [55].

In the present work, both the acute and chronic, life-long vascular effects of CB_1_R stimulation have been studied. With the aid of this genetically modified strain we were able to identify several functions of the ECS in controlling vascular contraction–relaxation and vascular wall remodeling.

We have found that vasodilatory effects were augmented in CB_1_R KO mice. Applying the specific inhibitors LNA and INDO, we have revealed that NO-dependent relaxations to Ach and estradiol have been augmented in CB_1_R KO mice and the observed presence of constrictor prostanoids in the aorta of wild-type mice has been missing in CB_1_R KO animals. Thus we have found that, in the absence of CB_1_Rs, the vasoregulatory role of constrictor prostanoids has disappeared while the NO-dependent effects have been augmented. In our previous studies we have observed that constrictor prostanoids were released together with the vasodilatory NO, their effect was reduced in trained animals [24,56].

In our set-up, the main vasorelaxant mediator was the NO pathway, the NO produced by the eNOS. We found significant eNOS expression upregulation in the CB_1_R deficient group compared to the control animals. On the other hand, Stanley found that cannabidiol (CBD) significantly increased phosphorylation of eNOS in endothelial cells, increasing their enzymatic activity [57]. The use of CB_1_ receptor antagonist AM251 inhibited CBD-induced vasorelaxation. Their results were confirmed by the use of CB_1_ receptor antagonist LY320135, which also significantly reduced CBD-induced vasorelaxation. The explanation can be that a lasting endocannabinoid effect through CB_1_Rs suppresses the expression of eNOS, while cannabinoid agonists and endothelial vasodilators such as Ach and estrogen activate the existing protein molecules. In addition, CBD can behave as an antagonist in the presence of more powerful stimulators of CB_1_Rs. Not to mention several other actions of CBD such as it being an agonist of PPARγ, TRPV-1 channels and also as having serotonin receptor stimulator activity [58].

Prostaglandins can mediate both constrictor and dilator responses in vascular tissue. In a recent study, Eckenstaler showed that pharmacological inhibition of COX-2 significantly reduced the synthesis of endothelial prostaglandins, whereas the synthesis of endothelial thromboxane (TXA_2_) was only partially reduced [59]. Endothelial cells produce COX metabolites including PGD_2_, PGE_2_, PGF_2α_, PGI_2_ and TXA_2_ targeting receptors on vascular smooth muscle, among them PGI_2_, PGD_2_ and PGE_2_ are vasodilators, PGF_2α_ and TXA_2_ are vasoconstrictors acting on FP/TP receptors [25].

Based on our results, we conclude that, although there is no difference in TP-R receptor density, the reduced COX-2 density and the tendency of reduced thromboxane synthase in CB_1_R KO mice may indicate that there is less TXA_2_ or some other vasoconstrictor prostanoid. Thus, we suggest that the proportion of constrictor prostanoids in the “prostanoid cocktail” is relatively reduced in the CB_1_R-deficient state. Also, based on our current experience, we hypothesize that the cannabinoid system may affect eNOS rather than prostacyclin-dependent vasorelaxant processes.

In our results, we found significant intima–media ratio reduction with media proliferation in the CB_1_R deficient group compared to the control animals. Is it an endocannabinoid-dependent action or not? Regarding Wang et al.’s study [60], our results can be partly controversial, because they described that the CB_1_ receptor deficiency of medial smooth muscle cells resulted in only limited effects on their contractility. However, in our case, the endothelial CB_1_ receptors were missing too.

We have found significantly reduced ER-β (estrogen receptor beta) density in the CB_1_R deficient group with preserved ER-α density. Arayan found that both estrogen receptors can modulate vascular smooth muscle cell (VSMC) proliferation [43]. Therefore, *our hypothesis can be that the ER-β decrease can influence the structural remodeling processes in the vascular wall.* On the other hand, our SMA (smooth muscle actin) immunohistochemical staining revealed no difference in smooth muscle contractile protein density, *knocking out the CB_1_R-induced smooth muscle proliferation that resulted in well differentiated cells* (cytoplasm filled with actin). Another observation of ours was that no difference in the density of elastic fibers was found between the two genetic groups, so the elastin secretion by the newly formed smooth muscle cells also remained unaltered. There is some contradiction with Molica’s study, who described that CB_1_R agonism promotes VSMC proliferation [61]. It is important that, in our study, we used healthy female mice while, in Molica’s study, the vessels underwent balloon catheter injury. Cannabinoid and endocannabinoid effects might be different in the normal and in the damaged vascular wall.

### 3.3. Role of ECS in Cardiovascular Pathologies and Vascular Remodeling, Possible Therapeutic Effects

In our present study we found that an absence of CB_1_ receptors improves Ach- and estradiol-induced vasodilations. In the absence of CB_1_R, NO-dependent vasodilations were augmented and constrictor prostanoid effects were diminished. Vascular histology revealed positive vascular remodeling processes: decreased intima/media ratio, improved NOS density and depressed COX density in CB_1_R KO mice. These indicate mostly beneficial endothelial and structural remodeling. Regarding cardiovascular diseases (CVDs), ECS may have an impact on the development of hypertension, atherosclerosis and on related diseases, including obesity and metabolic syndrome with dyslipidemia. This way, ECS signaling mechanisms and components may serve as potential targets of therapy [3,4,9,11]. Further, ECS may have a role in excessive hypotension, can have a protective-compensatory role in inflammatory and hypertensive diseases. Cannabinoids also have a local regulatory role through the modulation of the function of numerous ion channels [62]. It was found that inhibition of the degradation of endocannabinoids by FAAH inhibitor decreased the blood pressure of hypertensive rats [11]. According to an earlier observation in humans, 30 mg of THC elevated the blood pressure in normotensive healthy humans, however, an acute administration of an elevated dosage of THC (600 mg) caused hypotension. In normotensive rats, WIN 55,212-2 elevated the blood pressure, but in hypertensive rats, the same CB agonist reduced tension [63]. The explanation of the divergent observations can be a triphasic blood pressure effect of cannabinoids. The presence of the components of the ECS in so many tissues and cells of the body can induce such complicated effects [64].

Considering the potential role of ECS in the development of CVDs and vascular remodeling, stimulation and overexpression of CB_1_Rs can cause dyslipidemia and obesity, conditions leading to cardiovascular diseases. Administration of CB_1_R agonist elevates reactive oxygen species, inducing apoptosis of the endothelial cells of coronary arteries, while the CB_1_R antagonist rimonabant has been shown to decrease CVD risk and the development of atherosclerosis [65,66,67]. In our studies, we did not find any alteration in the nitrosylation of vascular wall proteins in CB_1_R KO animals (Figure 8A,B).

According to our observations, in the absence of CB_1_Rs, several functional and structural vascular parameters are altered in a manner which can be considered advantageous. This suggests that pharmacological modulation of the ECS, such as inhibiting CB_1_R or its signaling pathway, may form a promising therapy for several cardiovascular diseases.

### 3.4. Estrogen-Induced Relaxation, Plasma Levels, Receptors

During mass spectrometry, analysis of hormone levels in the plasma of CB_1_ receptor-deficient mice, we found a significantly higher level of conjugated estradiol in CB_1_R KO animals compared to the wild strain. The level of 4-hydroxyestrone was slightly elevated in CB_1_R KO female mice. Additionally, there was no significant difference observed in the levels of free estradiol. These results suggest that decreased functioning of the endocannabinoid system leads to increased estrogen conjugation and partially increased metabolism. The majority of estrogen compounds are present in conjugated form in circulation, with minimal presence of the free form. Thus these results clearly indicate that, in female CB_1_R KO mice, estrogen production is significantly increased, and there is a slight increase in the 4-hydroxy-metabolic pathway, likely due to the increased activity of the catalyzing enzyme CYP1B1, which can also be activated in numerous pathogenic processes (e.g., tumorigenesis) [68]. In our study, we found that the estradiol level was significantly elevated in the CB_1_R KO group. The ER-α density in the aortic wall was not different, while there was a reduction in ER-β. ER-α was described to trigger the NO-dependent pathway [69]. The role of vascular ER-β receptor is still unclear, the decreased expression we found can mark a compensatory negative feedback for this type of receptor as has been demonstrated for ER-α in other tissues [70]. The receptor feedback mechanism is still unclear. ER-β is known for its important role in the control of energy homeostasis [71] while the endocannabinoid system also plays an important role in the metabolic household [9,72]. This phenomenon further suggests an interplay between the estrogen and the endocannabinoid system in the regulation of the metabolism.

In previous studies, estrogen-induced vasorelaxation has been observed by complex acute nongenomic effects mediated by endothelial NO [26,27,28,69]. Vasodilatory effects of estradiol might contribute to maintaining cardiovascular protection in fertile women before menopause [29]. We found that estradiol-induced vasodilation has been augmented in CB_1_R KO mice. The vasodilatory effect of estradiol decreased due to eNOS inhibition with LNA in both groups, which result underlines the vasodilatory effect of estradiol through endothelial NO. The above mentioned effects are in line with results of other experts [27,69]. We also suggest from our results (Figure 2C,D) that, in females, the balance of constrictor and dilator prostanoids may be altered in the cannabinoid state: the balance of constrictor prostanoids may be dominant in vessels of WT compared with KO mice that would result in an augmented vasodilator response in CB_1_R KO [25,73]. It has some relevance with our observations that THC has an anti-estrogen effect by overwriting the nuclear ER-ß, and interferes with the activity of ER-α, and inhibits the gene expression regulated by estradiol and ER-α [74,75]. On the other hand, endocannabinoid AEA has been shown to stimulate the eNOS pathway and potentiate the effect of E2 [76].

### 3.5. Nitrative Stress Levels

Based on the eNOS results, we wanted to investigate the level of nitrative stress, which can modulate the NO bioavailability [77]. The endocannabinoids or exocannabinoids can modulate the level of oxidative–nitrative stress. The induced effects are controversial. THC administration can reduce the endotoxemic effect in a LPS (lipopolysaccharide) induced rat model [78], on the other hand, CB_1_ receptor activation can elevate apoptotic signals [77]. The reason for this Janus-face phenomenon can be that, in these studies, the cannabinoid system is examined in pathologic conditions. In our study, however, we investigated its function in healthy conditions without the activation of the inflammatory processes in the vascular wall. Based on our observations, the endocannabinoid system does not alter nitrative stress levels but modulates the NO household via its influence on the eNOS activity.

### 3.6. Gender Differences in Vascular Responses of Estrogens and Cannabinoids

Several animal and clinical studies have been performed to reveal gender differences in vascular functions. Estradiol via ER-α in vascular cells alters the expression and function of eNOS/NO, EDHF, ROS, PGI_2_, TXA_2_, it may alter the prostanoid balance. Previously a higher constrictor response was found for the thromboxane analogue in male coronary vessels compared to females [79]. In our study, the augmented relaxation response observed in female aortas in CB_1_R KO mice may also indicate that the balance of these mechanisms will be altered to augment vasodilation over constrictive effects [73]. Generally, in female mice, a higher relaxation (mainly NO mediated) and reduced contraction effects can be observed in isolated vessel studies [20]. There is a gender difference in the cannabinoid sensitivity based on different CB receptor expression and its influence on ovarian hormones estradiol and progesterone. In isolated mesenteric arteries, estradiol was shown to augment anandamide-induced relaxation, which was maximal in female hypertensive rats [80]. Our studies confirm that vascular effects of estradiol involve NO- and prostanoid-mediated mechanisms.

## 4. Materials and Methods

### 4.1. Chemicals

Phenylephrine, an adrenergic alpha receptor agonist, acetylcholine, an NO-dependent vasodilator, Angiotensin II, an AT_1_ receptor agonist, estradiol, WIN 55,212-2, a CB_1_ receptor synthetic agonist and the NOS inhibitor Nω-nitro-L-arginine as well as the cyclooxygenase inhibitor indomethacin were purchased from Merck KGaA. (Darmstadt, Germany). SQ 29,548, a TP receptor inhibitor, was purchased from Cayman Chemical (Ann Arbor, MI, USA, Appendix A). Standards for estrogen level determination, estron-(-,3,4-13C3) solution, 17beta-estradiol-D5 and all other salts and chemicals were purchased from Merck KGaA (Darmstadt, Germany). Solvents for E2, WIN 55,212-2 and indomethacin were diluted as stock solution in dimethyl sulfoxide (DMSO) and further diluted in Krebs solution on the day of the experiment. SQ 29,548 was diluted in ethanol as stock solution as suggested by the manufacturer. Similar dilution method with DMSO or ethanol was used as “vehicle”. LNA was diluted in Krebs solution with an ultrasound dispersion. Other solvents were diluted in Krebs from stock solution. Krebs and K-Krebs solutions were made on the day of the experiment from the following components: NaCl, KCl, CaCl_2_·2H_2_O, MgSO_4_·7H_2_O, NaHCO_3_, KH_2_PO_4_, EDTA, glucose (see Section 4.3). For a list and the sources of histological and immuno-histological reagents and their dilutions, see the corresponding subchapters.

### 4.2. Animals and Preparation

Female homozygous 4–6 months old (20–23 g) CB_1_R-knockout mice (CB_1_R KO, CB_1_R−/−, *n* = 25) [81] and wild-type counterpart mice (CB_1_R+/+; *n* = 35) were used. Animals were anaesthetized with pentobarbital (Euthasol, ASTfarma, Holland, 50 mg/kg) injected peritoneally, and an additional dose of Euthasol (5–10 mg/kg) was given if needed. After anesthesia, heparin was applied intraperitoneally and thoracotomy was performed. By cutting the vena cava inferior and superior we were able to obtain blood samples, anticoagulated with K3-Ethylenediamine-tetra-acetic acid (EDTA) in vacutainers (Vacuette tube 1 mL, Greiner Bio-One GmBH, Kremsmünter, Austria). After separation, plasma samples were stored at −80 °C for later estrogen level determination. Animals were perfused with saline through the left ventricle and the abdominal aorta was isolated under the preparation microscope (World Precision Instruments Inc., FL, USA-Experimetria LTD, Budapest, Hungary). Upper segments of abdominal aorta were isolated for immunohistochemistry. The investigations conformed the instructions of the Guide for the Care and Use of Laboratory Animals (NIH 8th Edn 2011), Institutional and National guidelines for animal care and were approved by the Animal Care Committee of the Semmelweis University, Budapest and by Hungarian authorities (No. PE/EA/1428-7/2018).

### 4.3. Myography

Abdominal aortic rings were subjected to wire myography tests as described before [5,82]. An isolated segment was put into cold Krebs solution containing (in millimolar): 119 NaCl, 4.7 KCl, 2.5 CaCl_2_·2H_2_O, 1.17 MgSO_4_·7H_2_O, 20 NaHCO_3_, 1.18 KH_2_PO_4_, 0.027 EDTA, 10.5 glucose. The 3–4 mm wide aortic rings were mounted on the holders of the multichamber isometric myograph system (610 M Multiwire Myograph System, Danish Myo Technology A/S, Aarhus, Denmark) to record isometric tension. Eight channels were examined and recorded simultaneously by the Powerlab data acquisition system, evaluated later with the LabChart version 8 evaluation software (ADInstruments, Oxford, UK-Ballagi LTD, Budapest, Hungary). The myograph chambers were filled with Krebs solution, were thermostated at 36 °C and bubbled with carbogenic gas (95% O_2_ + 5% CO_2_), holding the pH at 7.4. According to our protocols [5], abdominal aortic segments were prestreched to 10 mN and were allowed to equilibrate for 30 min. After the equilibration period, reference contraction was performed with hyperkalemic Krebs solution (K^+^ 124 mmol/L). Concentration–response curves to vasoconstrictor Ang II (1–100 nmol/L) and phenylephrine (1 nmol/L–10 µmol/L) and to the vasodilator acetylcholine (1 nmol/L–1 µmol/L) after preconstriction with phenylephrine were obtained in order to test smooth muscle and endothelium functions. Selective inhibitors were applied to test the mechanism of endothelial function: nitric oxide synthase was inhibited by Nω-nitro-L-arginine, COX was inhibited by indomethacin, parallel segments were treated with vehiculum as controls. Drugs were applied for 20 min prior the repetition of the dose–response curves (Phe and Ach). Estradiol relaxation (10 nmol/L–10 µmol/L) was tested without and with the presence of inhibitors in parallel segments after submaximal preconstriction with Phe (5 µmol/L). Vasoconstriction responses were normalized to KCl contraction, vasodilation responses were calculated compared with the precontraction state [5,82]. Effects of CB_1_ receptor activation with synthetic cannabinoid receptor agonist (WIN 55,212-2, 10 µmol/L) in both WT (CB_1_R +/+) and CB_1_R KO (CB_1_R−/−) groups were tested. WIN 55,212-2-induced relaxation was also compared against “vehicle”, in which relaxation effect was not observed. Effects of inhibitors were also calculated as changes in contraction force and as attenuation in relaxation (differences from control, Figure 3A–F) and also as relative (%) changes from control (difference from control/control levelx100, Figure 3G). Additional experiments were performed with TP receptor inhibitor SQ 29,548 in which Ach-induced relaxations were repeated (Appendix A).

### 4.4. Hormone Determinations

Estrogens and estrogen metabolites: free estradiol, conjugated estradiol, free and conjugated estrone, 2-hydroxyestrone and 4-hydroxyestrone levels were determined by liquid chromatography-tandem mass spectrometry from blood plasma samples. Free and conjugated levels of estrone, 2-hydroxyestrone and conjugated 4-hydroxyestrone were under threshold levels (see also Appendix A).

### 4.5. Histological and Immuno-Histochemical Stainings

Paraformaldehyde (PFA) fixed, paraffin embedded abdominal aortic sections, 7 µm thick, were cut. Hematoxylin eosin staining (HE) was used for topography analysis, e.g., intima–media ratio, and to examine morphological changes. Incubation with hematoxylin (Hematoxylin modified to Gill II, Sigma-Aldrich, St. Louis, MO, USA; Eosin Y, Merck Millipore, Burlington, MA, USA).

Elastic fibers were stained with resorcin fuchsin (Electronmicroscopy Sciences, Hatfield, PA, USA).

Sections were immuno-histochemically stained against α-smooth muscle actin endothelial nitric oxide synthase, cyclooxygenase-2, TP receptor, estrogen receptor α and β (ER-α; ER-β) and 3-nitrotyrosine (NT). After deparaffinization, antigen retrieval was performed by heating the slides in citrate buffer (pH = 6) in case of α-SMA; eNOS, COX-2 and ER-β. Proteins were digested with Proteinase K (1 mg/mL in phosphate buffer (Merck Millipore, Burlington, MA, USA) for ER-α and NT stainings. Endogenous peroxidase activity was blocked with 3% H_2_O_2_. To eliminate the nonspecific labelling of the secondary antibody, we used a 2.5% normal horse serum (NHS) blocking solution (Vector Biolabs, Burlingame, CA, USA).

Additional experiments were performed to detect thromboxane synthase (Appendix A).

Primary antibodies used with overnight application at 4 °C were as follows: α-SMA mouse monoclonal antibody 1:10,000 (Abcam, Cambridge, UK); eNOS mouse monoclonal antibody 1:50 (Abcam, Cambridge, UK); COX-2 rabbit polyclonal antibody 1:200 (Abcam Cambridge, UK); ER-β rabbit polyclonal antibody 1:500 (Invitrogen Waltham, MA, USA); TP rabbit polyclonal antibody 1:50 (MyBioSource, San Diego, CA, USA); ER-α rabbit polyclonal antibody 1:80 (Merck Millipore, Burlington, MA, USA); NT rabbit polyclonal antibody 1:250 (Merck Millipore, Burlington, MA, USA). For secondary labeling we used horseradish-peroxidase-(HRP) linked anti-mouse (in case of α-SMA and eNOS stainings) or anti-rabbit (in case of ER-α, ER-β, COX-2; TP-R and NT stainings) IgG polyclonal antibodies (Vector Biolabs, Burlingame, CA, USA). Visualization was performed by brown colored horse radish peroxidase (HRP) linked 3′3-diaminobenzidine (DAB, Vector Biolabs, Burlingame, CA, USA). For counterstaining we used purple colored hematoxylin (Hematoxylin modified to Gill II, Sigma-Aldrich, St. Louis, MO, USA).

After covering slides were photographed with Nikon eclipse Ni-U microscope with DS-Ri2 camera (Nikon, Minato—Tokyo, Japan). Photos of the eNOS, COX-2 and ER-α slides were taken at 20× magnification, in other cases (including HE and RF stainings) we used 10× magnification. On HE slides the thickness of the aortic wall, and separately the thickness of the intimal and medial layers, were measured. Noncalibrated optical density (O.D.) on the RF stained slides characterized the elastic density. On immunohistochemical slides, the brown positivity and the background staining (DAB + hematoxylin) were separated, staining intensity was determined by noncalibrated optical density. In the case of eNOS and COX-2, only the endothelial layer was evaluated, while in the case of ER-α and ER-β both the endothelial and the medial layers were evaluated separately. In the remaining cases we investigated the staining intensity in the medial layer using the FIJI^®^ software (https://imagej.net/software/fiji/downloads, National Institutes of Health, Bethesda, MA, USA).

### 4.6. Statistical Analysis

In wire myography experiments, contraction data were normalized to KCl contraction (taken as 100%), relaxation data were calculated as percent values of precontraction level. Statistical analysis was performed with two-way ANOVA and Bonferroni post hoc test for the analysis of comparisons between the two groups. Unpaired *t*-test was applied for comparison between groups. Histological and immunohistochemical results were tested with an unpaired *t*-test. Values were expressed as mean ± standard error of mean (mean ± SEM), significance threshold was set at *p* < 0.05, Analysis was performed by the SigmaStat software 3.5s (Systat Software Inc., San Jose, CA, USA) and with GraphPad PRISM 9.5.0. (San Diego, CA, USA). Pharmacological parameters of EC50, Emax and pEC50 were also analyzed with inhibitor data of Figure 2 (Appendix A).

## 5. Conclusions

We have found a functional and structural vascular remodeling of the rodent aortic wall in the absence of cannabinoid receptors type 1 (CB_1_R KO) indicating improved vasodilation responses partially mediated by nitric oxide and the alteration of vasoactive endoprostanoid function. In CB_1_R KO mice, the relaxation responses are free from the dominance of constrictor prostanoids, which are present in WT. Our histology and immunohistochemistry results demonstrated a lower intima–media ratio in the CB_1_R KO group together with lower COX-2 and higher eNOS densities in accordance with their functional results. We also found a significant reduction in ER β receptor density in the CB_1_ receptor deficient group, which may be related to the elevated estrogen plasma levels detected in the receptor deficient group, suggesting a potential negative feedback mechanism. Our results on genetically modified animals reveal the functional role of CB_1_R throughout the lifecycle. We have a good reason to think that a substantial part of the observed effects can be elicited by chronic administration of receptor inhibitors. Our observations give further support to earlier views that the suppression of the endocannabinoid system by the inhibition of CB_1_Rs may exert beneficial vascular effects by improving vasodilation and promoting favorable vascular remodeling.

## Figures and Tables

**Figure 1 ijms-24-16429-f001:**
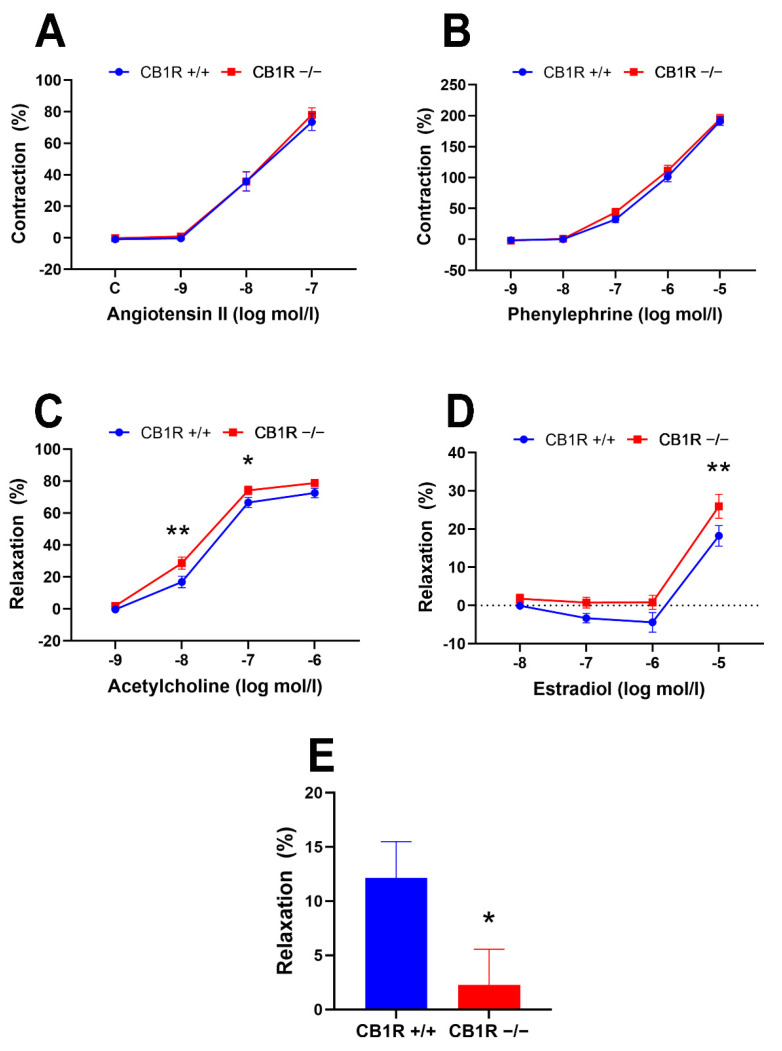
Contraction–relaxation vascular responses of aortic segments of wild-type (CB1R+/+) and CB1R knockout (CB1R−/−) female mice. **Panel** (**A**). Dose–response contraction curves to angiotensin II in wild-type (*n* = 14–25 rings from 9 animals) and in CB1R knockout (18–26 segments from 9 animals). **Panel** (**B**). Dose–response contraction curves to phenylephrine in CB1R+/+ (*n* = 28 segments from 9 animals) and in CB1R−/− (*n* = 27 segments from 9 animals). **Panel** (**C**). Dose–response relaxation curves to acetylcholine in CB1R+/+ (*n* = 26 segments from 9 animals) and in CB1R−/− (*n* = 27 segments from 8 animals). **Panel** (**D**). Dose–response relaxation curves to estradiol in CB1R+/+ (*n* = 8–10 segments from 8 animals) and CB1R−/− (*n* = 7–10 segments from 10 animals). **Panel** (**E**). Relaxation response of aortic segments to the CB1R agonist WIN 55,212-2 (10µM) in CB1R+/+ (*n* = 5) and CB1R−/− (*n* = 6) female mice. Mean ± SEM values. *p* < 0.05 values were considered significant. *: *p* < 0.05 and **: *p* < 0.01 between wild-type (CB1R+/+) and CB1R knockout (CB1R−/−) groups in two-way ANOVA with a Bonferroni post hoc test and unpaired *t*-test (panel E) (Contraction data were normalized to KCl contraction, relaxation data were calculated as percent values of precontraction level).

**Figure 2 ijms-24-16429-f002:**
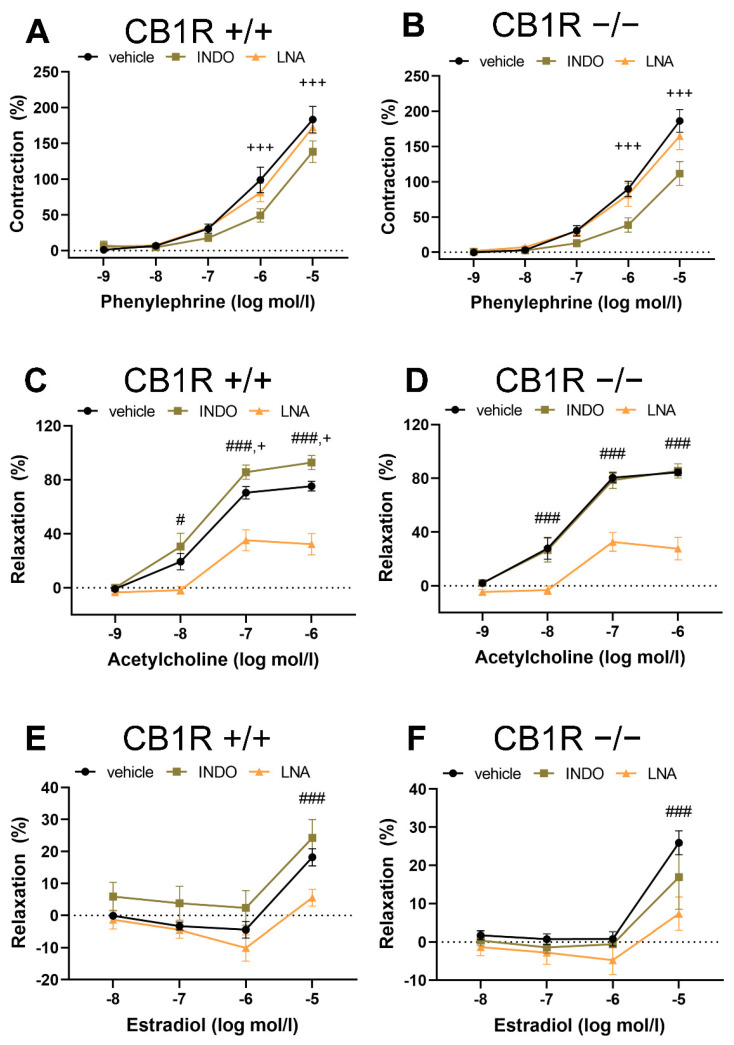
Effects of specific inhibitors Nω-nitro-L-arginine (LNA, inhibitor of nitric oxide synthase) and indomethacin (INDO, inhibitor of cyclooxygenase) on contraction–relaxation vascular responses in aortas of wild-type (CB1R+/+) and CB1R knockout (CB1R−/−) female mice. **Panel** (**A**). Effects of inhibitors on phenylephrine-induced vasoconstriction in CB1R+/+ female mice (*n* = 7–9, segments = 7–9). **Panel** (**B**). Effects of inhibitors on phenylephrine-induced vasoconstriction in CB1R−/− female mice (*n* = 8, segments = 8). **Panel** (**C**). Effects of inhibitors on acetylcholine-induced vasodilation in CB1R+/+ female mice (*n* = 8–9, segments = 8–9). **Panel** (**D**). Effects of inhibitors on acetylcholine-induced vasodilation in CB1R−/− female mice (*n* = 8–9, segments = 8–9). **Panel** (**E**). Effects of inhibitors on estradiol-induced vascular responses in CB1R+/+ female mice (*n* = 10, segments = 7–10). **Panel** (**F**). Effects of inhibitors on estradiol-induced vascular responses in CB1R−/− female mice (*n* = 10, segments = 7–10). Data are shown as mean ± SEM values. *p* < 0.05 values were considered significant. #: *p* < 0.05, ###: *p* < 0.001 between vehicle and LNA-treated segments, +: *p* < 0.05, +++: *p* < 0.001 between vehicle and INDO-treated segments (two-way ANOVA with a Bonferroni post hoc test). Abbreviations: INDO: indomethacin, LNA: Nω-nitro-L-arginine, CB1R: cannabinoid type 1 receptor, CB1+/+: CB1R wild-type, CB1−/−: CB1R knockout mice (Contraction data were normalized to KCl contraction, relaxation data were calculated as percent values of precontraction level).

**Figure 3 ijms-24-16429-f003:**
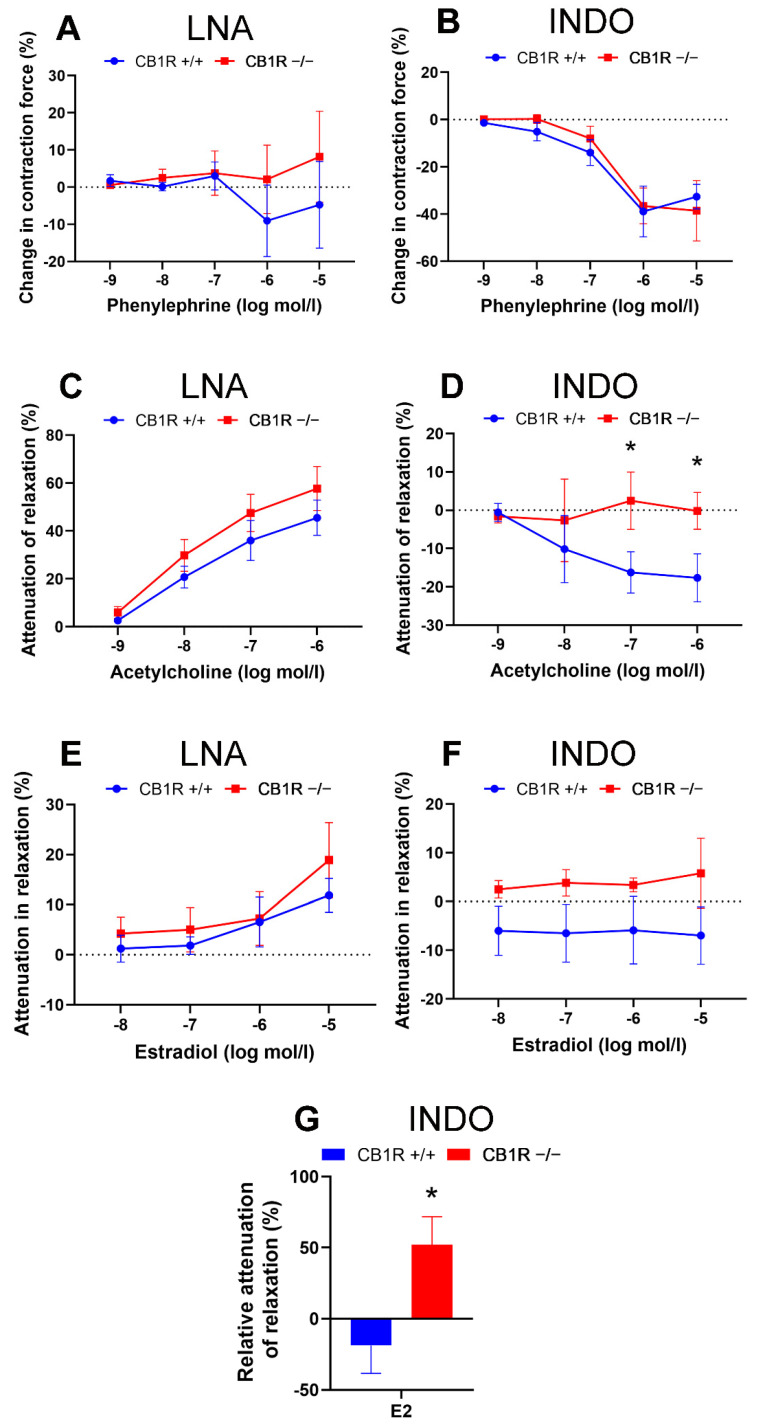
Comparison of the effects of specific inhibitors Nω-nitro-L-arginine (LNA, inhibitor of nitric oxide synthase) and indomethacin (INDO, inhibitor of cyclooxygenase) on contraction-relaxation vascular responses in aortas of WT (CB1R+/+) and CB1R KO (CB1R−/−) female mice. **Panel** (**A**). Effects of inhibitor LNA on Phe-induced vasoconstriction in CB1R+/+ (*n* = 6, segments = 6) and CB1R−/− (*n* = 6, segments = 6) female mice. **Panel** (**B**). Effects of inhibitor INDO on Phe-induced vasoconstriction in CB1R+/+ (*n* = 5, segments = 5) and CB1R−/− (*n* = 6, segments = 6) female mice. **Panel** (**C**). Effects of inhibitor LNA on Ach-induced vasorelaxation in CB1R+/+ (*n* = 8, segments = 8) and CB1R−/− (*n* = 8, segments = 8) female mice. **Panel** (**D**). Effects of inhibitor INDO on Ach-induced vasodilation in CB1R+/+ (*n* = 8, segments = 8) and CB1R−/− (*n* = 8, segments = 8) female mice. **Panel** (**E**). Effects of inhibitor LNA on E2 induced vascular responses in CB1R+/+ (*n* = 9) and CB1R−/− (*n* = 8) female mice. **Panel** (**F**). Effects of inhibitor INDO on E2-induced vascular responses in CB1R+/+ (*n* = 9) and CB1R−/− (*n* = 9) female mice. Differences in contraction or relaxation values are plotted. **Panel** (**G**). Normalized effects of INDO on E2 relaxation responses (at 10 µmol/L). Values are plotted as percentage differences from control relaxation. Mean ± SEM values. *p* < 0.05 values were considered significant. * *p* < 0.05 between CB1R+/+ and CB1R−/− groups (two-way ANOVA with a Bonferroni post hoc test). Abbreviations: Ach: acetylcholine, Phe: phenylephrine, E2: estradiol, INDO: indomethacin, LNA: Nω-nitro-L-arginine, CB1R: cannabinoid type 1 receptor, WT: wild-type, KO: knockout, CB1R+/+: cannabinoid type 1 receptor wild-type mice, CB1R−/−: cannabinoid type 1 receptor knockout mice.

**Figure 4 ijms-24-16429-f004:**
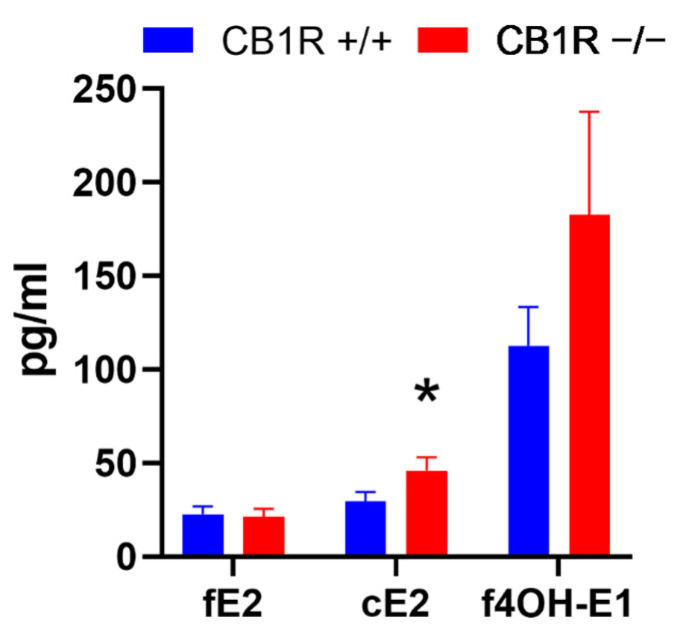
Free estradiol, conjugated estradiol and free 4-hydroxyestrone levels (pg/mL) in WT and CB1R KO female mice. Mean ± SEM values. *p* < 0.05 values were considered significant. *: *p* < 0.05 between CB1R+/+ (*n* = 14–24) and CB1R−/− (*n* = 11–17) groups. Abbreviations: fE2: free estradiol level, cE2: conjugated estradiol level, f4OH-E1: free 4-hydroxyestrone level, CB1R: cannabinoid type 1 receptor, WT: wild-type, KO: knockout, CB1R+/+: cannabinoid type 1 receptor wild-type mice, CB1R−/−: cannabinoid type 1 receptor knockout mice. *: *p* < 0.05 between groups (unpaired *t*-test). Note that sample takings were not adjusted to the estrus cycle, mean values reflect means during estrus cycle.

**Figure 5 ijms-24-16429-f005:**
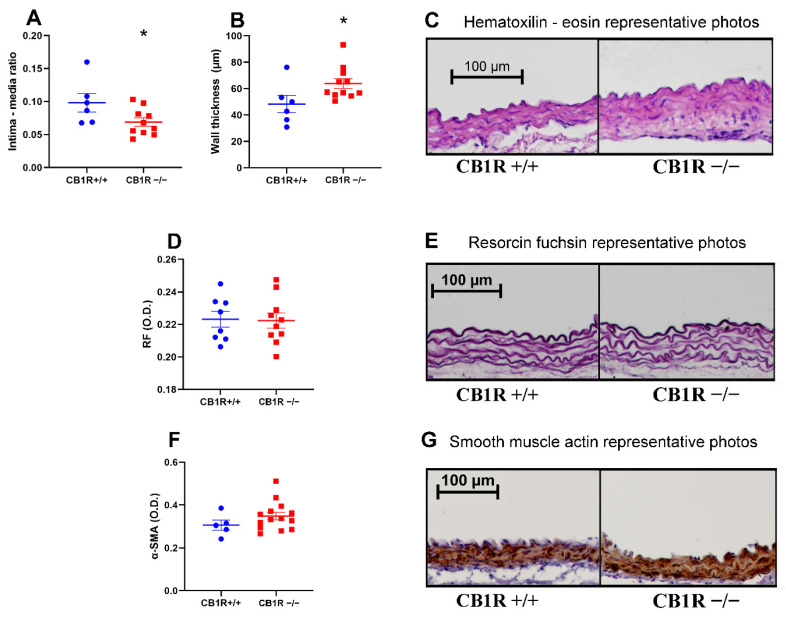
Morphological changes of the abdominal aorta wall. **Panel** (**A**). Intima–media ratio of the aorta wall with mean ± SEM, *n* = 6 (CB1R+/+) *n* = 10 (CB1R−/−) *: *p* < 0,05 CB1R+/+ vs. CB1R−/− group. **Panel** (**B**). Aorta wall thickness in micrometer with mean ± SEM *n* = 6 (CB1R+/+) *n* = 11 (CB1R−/−) *: *p* < 0.05 CB1R+/+ vs. CB1R−/− group. **Panel** (**C**). Representative photos of the hematoxylin-eosin staining photographed at 10× magnification. **Panel** (**D**). Optical density of elastic fiber on resorcin fuchsin stained sections. *n* = 7 (CB1R+/+), *n* = 10 (CB1R−/−). **Panel** (**E**). Representative photos of the resorcin fuchsin stained sections photographed at 10× magnification. **Panel** (**F**). Optical density of α-SMA stained sections. *n* = 5 (CB1R+/+) *n* = 14 (CB1R−/−) **Panel** (**G**). Representative photos of α-SMA stained aorta segments, visualization with diamino-benzidine (DAB) on hematoxylin counterstaining, photographed at 10× magnification. Statistical analysis executed with unpaired *t*-test. Data shown by noncalibrated optical density with mean ± SEM Abbreviations: CB1R: cannabinoid type 1 receptor, CB1R+/+: cannabinoid type 1 receptor wild-type mice, CB1R−/−: cannabinoid type 1 receptor knockout mice. RF: resorcin fuchsin, α-SMA: alpha smooth muscle actin, O.D.: optical density.

**Figure 6 ijms-24-16429-f006:**
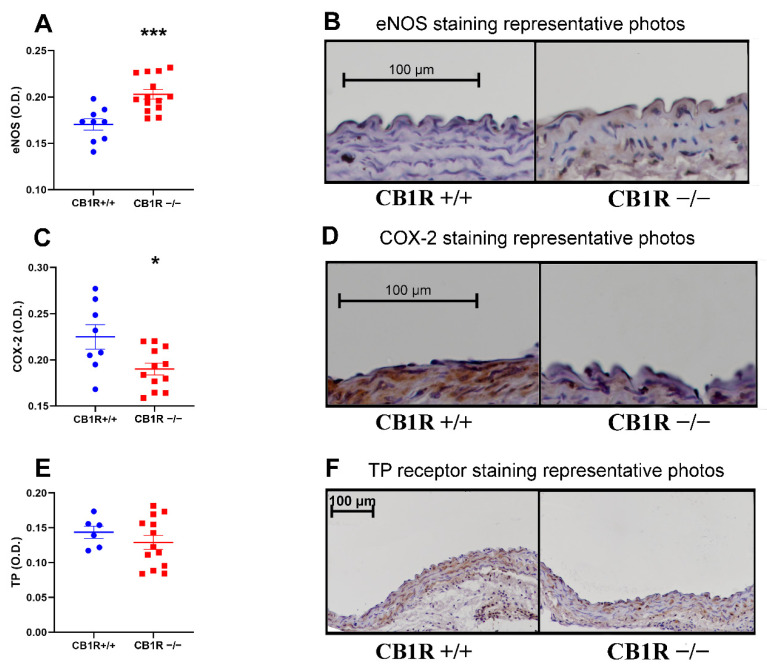
Vasoactive markers of the abdominal aorta wall. **Panel** (**A**). Results of eNOS immunostained sections. *n* = 9 (CB1R+/+) *n* = 14 (CB1R−/−). ***: *p* < 0.001 CB1R+/+ vs. CB1R−/− group. **Panel** (**B**). Representative photos of eNOS immunostained aorta segments, visualization with DAB on hematoxylin counterstaining, photographed at 20× magnification. Evaluation performed from the values of the endothelial layer **Panel** (**C**). Results of COX-2 immunostained sections. *n* = 8 (CB1R+/+), *n* = 12 (CB1R−/−). *: *p* < 0,05 CB1R+/+ vs. CB1R−/− group. **Panel** (**D**). Representative photos of COX-2 immunostained aorta segments, visualization with DAB on hematoxylin counterstaining, photographed by 20× magnification. Evaluation performed from the values of the endothelial layer **Panel** (**E**). Results of TP receptor immunostained sections. *n* = 6 (CB1R+/+), *n* = 13 (CB1R−/−). **Panel** (**F**). Representative photos of TP receptor immunostained aorta segments, visualization with DAB on hematoxylin counterstaining, photographed at 10× magnification. Evaluation performed from the values of the media layer. Statistical analysis performed with unpaired *t*-test. Data shown by noncalibrated optical density with mean ± SEM Abbreviations: CB1R: cannabinoid type 1 receptor, CB1R+/+: cannabinoid type 1 receptor wild-type mice, CB1R−/−: cannabinoid type 1 receptor knockout mice, eNOS: endothelial nitric oxide synthase, DAB: diamino-benzidine, COX: cyclooxygenase, TP: thromboxane-prostanoid receptor O.D.: optical density.

**Figure 7 ijms-24-16429-f007:**
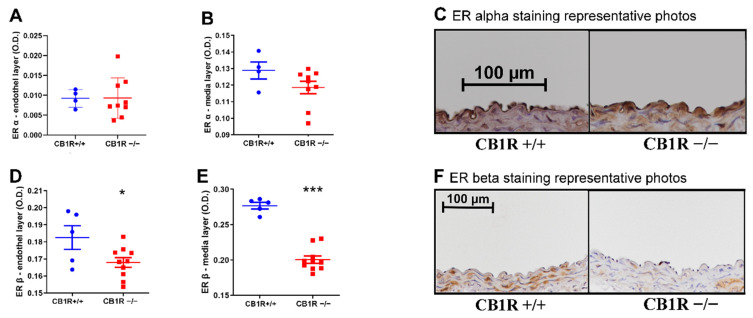
Estrogen receptor protein expression in the abdominal aorta wall. Panels (A, B) Results of ER-α immunostained sections. **Panel** (**A**). Results from the endothelial layer. *n* = 6 (CB1R+/+), *n* = 9 (CB1R−/−). **Panel** (**B**). Results from the media layer. *n* = 4 (CB1+/+) *n* = 9 (CB1R−/−). **Panel** (**C**). Representative photos of ER-α immunostained aorta segments, visualization with DAB on hematoxylin counterstaining, photographed at 20× magnification. Evaluation performed from the values of the endothelial and the media layer. **Panels** (**D**,**E**). Results of ER-β stained sections. **Panel** (**D**). Results from the endothelial layer. (CB1R+/+), *n* = 9 (CB1R−/−). *: *p* < 0,05 CB1R+/+ vs. CB1R−/− group. **Panel** (**E**). Results from the medial layer. *n* = 9 (CB1R+/+), *n* = 10 (CB1R−/−). ***: *p* < 0.001 CB1R+/+ vs. CB1R−/− group. **Panel** (**F**). Representative photos of ER-β immunostained aorta segments, visualization with DAB on hematoxylin counterstaining, photographed by 10× magnification. Evaluation performed from the values of the endothelial and the media layer. Statistical analysis performed with unpaired *t*-test. Data shown by noncallibrated optical density with mean ± SEM. Abbreviations: CB1R: cannabinoid type 1 receptor, CB1R+/+: cannabinoid type 1 receptor wild-type mice, CB1R−/−: cannabinoid type 1 receptor knockout mice, eNOS: endothelial nitric oxide synthase, DAB: diamino-benzidine, COX: cyclooxygenase, O.D.: optical density, ER: estrogen receptor.

**Figure 8 ijms-24-16429-f008:**
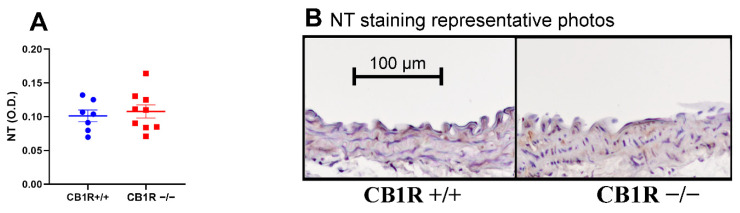
Results of 3-nitrotyrosine density in the abdominal aorta wall. **Panel** (**A**). Noncalibrated optical density data are shown with mean ± SEM, *n* = 7 (CB1R+/+), *n* = 9 (CB1R−/−). Statistical analysis performed with unpaired *t*-test. **Panel** (**B**). Representative pictures of 3-nitrotyrosine staining. Visualization with DAB on hematoxylin counterstaining, photographed at 10× magnification. Evaluation performed from the values of the media layer. Abbreviations: CB1R: cannabinoid type 1 receptor, CB1R+/+: cannabinoid type 1 receptor wild-type mice, CB1R−/−: cannabinoid type 1 receptor knockout mice, DAB: diamino-benzidine, O.D.: optical density, NT: 3-nitrotyrosine staining.

## Data Availability

Data are available in the Appendix A.

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
