# Peer review of "Role of CB1 Cannabinoid Receptors in Vascular Responses and Vascular Remodeling of the Aorta in Female Mice"

_ijms, 2023, doi:10.3390/ijms242216429_

Round 1

Reviewer 1 Report

Comments and Suggestions for Authors

Bányai and co-workers decided to determine the role of cannabinoid CB1 receptors in vascular responses and vascular remodeling of the aorta in female mice. For this purpose, they performed the experiments on valuable CB1R knockout (CB1R-KO), and wild-type (WT) female mice.

I have major concerns about the manuscript (mainly Introduction and Discussion), and my feedback is outlined below:

1.       Introduction and Discussion sections are based on too old positions (10, 32, 23, and 5 from <2000; 2001-2010, 2010-2020, and 2021-2023, respectively);

2.       The text regarding vascular cannabinoid receptors and their interactions with other vasoactive substances (mainly vasoconstrictors) is based mainly on the Authors’ own publications. Unfortunately, other relevant publications are missing; below only few examples:

-      Baranowska-Kuczko M, Kozłowska H, Kloza M, Harasim-Symbor E, Biernacki M, Kasacka I, Malinowska B. Beneficial changes in rat vascular endocannabinoid system in primary hypertension and under treatment with chronic inhibition of fatty acid amide hydrolase by URB597. Int J Mol Sci. 2021;22:4833. (describing, among others, the effects of CB1R antagonist on vasoconstriction of mesenteric artery and aorta induced by phenylephrine or thromboxane A2 analogue U46619 in spontaneously hypertensive rats and their normotensive controls, and levels of AEA and 2-AG in both above vessels);

-      Karpińska O, Baranowska-Kuczko M, Kloza M, Kozłowska H. Endocannabinoids modulate Gq/11 protein-coupled receptor agonist-induced vasoconstriction via a negative feedback mechanism. J Pharm Pharmacol. 2018;70:214-222. (the authors of this review refer to other relevant publications);

-      Ho WS. Modulation by 17β-estradiol of anandamide vasorelaxation in normotensive and hypertensive rats: a role for TRPV1 but not fatty acid amide hydrolase. Eur J Pharmacol. 2013;701:49-56. (interaction between 17β-estradiol and anandamide)

-      Szabó R, Börzsei D, Szabó Z, Hoffmann A, Zupkó I, Priksz D, Kupai K, Varga C, Pósa A. A potential involvement of anandamide in the modulation of HO/NOS systems: women, menopause, and "medical cannabinoids". Int J Mol Sci. 2020;21:8801.

3.   Authors did not discuss their current results in light of their previous results obtained on male CB1R-KO mice (also in aorta), in which they obtained the opposite effects to the currently described, e.g., in Szekeres et al. (2012) one can read that: “Ang II stimulates vascular endocannabinoid formation, which attenuates its vasoconstrictor effect, suggesting that endocannabinoid release from the vascular wall and CB1R activation reduces the vasoconstrictor and hypertensive effects of Ang II.” In the current manuscript one can find that: “endocannabinoids acting through CB1Rs control NOS expression in the aortic wall, moderating the amount of available enzyme molecules and reducing the extent of endothelial dilation”.

4.   The scientific hypothesis described by the Authors in the part 3.1 is interesting. However, it should be proven using additional pharmacological tools, such as TP receptor antagonists or inhibitors of endocannabinoid degradation. Only one example: how can Authors be sure that “constrictor prostanoids contribute to the vascular effects in WT but not in CB1R-KO mice” without the determination of their levels or without using specific antagonists of respective receptors? Especially, Authors did not find any effects of CB1R antagonists on the contraction of mouse aorta induced by PGF2α (Szekeres et al., 2015); this effect was also not mentioned in the current manuscript).

5.   Part 3.3 is too general, especially in light of very modest effects observed in the current study (e.g., relaxation by WIN55212-2 by about 12%), the examination of the conduit aorta but not resistance artery (e.g., mesenteric), and the fact that vascular CB1R does not play a central role in basal vascular health maintenance (e.g., Wang et al., 2022). This part seems like a very short review.

Minor comments: 

1.     The solvents for all compounds used in functional studies should be given in chapter 4.1. Moreover, the vascular effects of solvents should be examined. For example, WIN55212-2 requires other solvent than saline solution or water. It caused small relaxation by about 12%. The effect of the solvent is very important in this case.

2.     Do antagonists/inhibitors affect basal tension by themselves? The detailed values of basal tone in the presence of antagonists/inhibitors should be given in the manuscript. For example, the NOS inhibitor often increases basal tone.

3.     Statistical analysis – please verify in the Text and in the Legends, and justify the use of two-way ANOVA with Bonferroni post hoc test. It was also listed in the case of comparison of two groups only, e.g., in Fig. 1.

4.     All legends are too long. Authors should try to describe the whole Figures but not particular panels in the frame of the same figure. However, the meaning % in the case of contraction and relaxation should be explained in Legends.

Author Response

Answers for Reviewer 1

We thank for our Reviewer for the valuable, critical and supportive review of our manuscript and we believe that the corrections performed by the suggestions have significantly improved the quality of our manuscript.

We have answered point by point according to the suggestions and we have implemented them in the text.

Bányai and co-workers decided to determine the role of cannabinoid CB1 receptors in vascular responses and vascular remodeling of the aorta in female mice. For this purpose, they performed the experiments on valuable CB1R knockout (CB1R-KO), and wild-type (WT) female mice.

I have major concerns about the manuscript (mainly Introduction and Discussion), and my feedback is outlined below:

  1. Introduction and Discussion sections are based on too old positions (10, 32, 23, and 5 from <2000; 2001-2010, 2010-2020, and 2021-2023, respectively);

Thank you for the suggestion. We have added new citations to the paper, mostly recent publications, some of the suggested by the Reviewer. These papers are: Zhou et al. 2021, Afshar et al., 2022, Baranowska-Kuczko et al. 2021, Karpinska et al. 2018, Savva et al. 2020, Iorga et al. 2018, Aryan et al. 2020, Bondarenko et al. 2019, Pabbidi et al. 2018, Santoro et al. 2021, Szabo et al. 2020, Bányai et al. 2023, Valee et al. 2023, Eckenstaler et al., 2022, see refs: 15,18,25,31,41,42,43,55,59,62,73,74,76,78.

  1. The text regarding vascular cannabinoid receptors and their interactions with other vasoactive substances (mainly vasoconstrictors) is based mainly on the Authors’ own publications. Unfortunately, other relevant publications are missing; below only few examples:

-      Baranowska-Kuczko M, Kozłowska H, Kloza M, Harasim-Symbor E, Biernacki M, Kasacka I, Malinowska B. Beneficial changes in rat vascular endocannabinoid system in primary hypertension and under treatment with chronic inhibition of fatty acid amide hydrolase by URB597. Int J Mol Sci. 2021;22:4833. (describing, among others, the effects of CB1R antagonist on vasoconstriction of mesenteric artery and aorta induced by phenylephrine or thromboxane A2 analogue U46619 in spontaneously hypertensive rats and their normotensive controls, and levels of AEA and 2-AG in both above vessels);

-      Karpińska O, Baranowska-Kuczko M, Kloza M, Kozłowska H. Endocannabinoids modulate Gq/11 protein-coupled receptor agonist-induced vasoconstriction via a negative feedback mechanism. J Pharm Pharmacol. 2018;70:214-222. (the authors of this review refer to other relevant publications);

-      Ho WS. Modulation by 17β-estradiol of anandamide vasorelaxation in normotensive and hypertensive rats: a role for TRPV1 but not fatty acid amide hydrolase. Eur J Pharmacol. 2013;701:49-56. (interaction between 17β-estradiol and anandamide)

-      Szabó R, Börzsei D, Szabó Z, Hoffmann A, Zupkó I, Priksz D, Kupai K, Varga C, Pósa A. A potential involvement of anandamide in the modulation of HO/NOS systems: women, menopause, and "medical cannabinoids". Int J Mol Sci. 2020;21:8801.

We thank for our Reviewer for the valuable suggestions to update the reference list. We have cited articles based on the suggestions in the text. References: 15, 55, 76, 79 in the text in lines: 82,404,539,566.

  1. Authors did not discuss their current results in light of their previous results obtained on male CB1R-KO mice (also in aorta), in which they obtained the opposite effects to the currently described, e.g., in Szekeres et al. (2012) one can read that: “Ang II stimulates vascular endocannabinoid formation, which attenuates its vasoconstrictor effect, suggesting that endocannabinoid release from the vascular wall and CB1R activation reduces the vasoconstrictor and hypertensive effects of Ang II.” In the current manuscript one can find that: “endocannabinoids acting through CB1Rs control NOS expression in the aortic wall, moderating the amount of available enzyme molecules and reducing the extent of endothelial dilation”.

We agree with our Reviewer that our previous observations on male arteries (Szekeres et al. 2012 and Szekeres et al. 2015, refs 5,14) indicate that contractile effects e.g. to Ang II is augmented by the inhibition of CB1R which indicate an acute dilatative effect in the presence of CB1Rs. This effect is due to a calcium signaling-induced DAG lipase activation and endocannabinoid release. However, in the female aortas we could not demonstrate a significant functional effect of this mechanism when comparing contractile effects between WT and CB1R KO mice, which we attribute to the higher estrogen-induced influence to improve vasorelaxation effects and exert cardioprotection in females before menopause (Pabbidi  et al. 2018, ref 73). Generally, in female mice a higher relaxation (mainly NO-mediated) and reduced contraction effects can be observed in isolated vessel studies (Mericli M et al. 2004, A Huang et al. 1998, refs 20,26.) We believe that the acute and chronic effects of endocannabinoids on the vascular wall can affect endothelial vasodilation in the opposite direction.

Estradiol via ER alpha in vascular cells alters expression, production and function of eNOS/NO, EDHF, ROS, PGI2, TXA2 etc that may alter also prostanoid balance. In our study the augmented relaxation response observed in female aortas in CB1R KO mice may also indicate that the balance of these mechanisms will be altered to augment vasodilatation over constrictive effects (Pabbidi  et al. 2018 ref 73)

Related to the relaxation effect that is in reduced extent in female aortas compared to males we may propose by Pabbidi et al. and according to our results (Fig 2C-D) that in females the balance of constrictor and dilator prostanoids may be altered by the cannabinoid state: The balance to constrictor plostanoids may be dominant in vessels of WT compared KO mice that would result in and augmented vasodilator response in CB1R KO.

There is a gender difference in the cannabinoid sensitivity based on different CB receptors expression and influence on ovarian hormones estradiol and progesterone. Preclinical evidence supporting direct interactions between sex hormones and the endocannabinoid system may translate to sex differences in response to cannabis and cannabinoid use in men and women (Blanton et al. Sex differences and the endocannabinoid system in pain. Pharmacol Biochem Behav.2021 Mar;202:173107.)

Based on the suggestion of our Reviewer, we have added these explanations to the Discussion section, highlighting gender differences in the context of CB1R and vascular responses in Section 3.6. in Discussion and some explanation in Sections 3.2. and 3.4.

  1. The scientific hypothesis described by the Authors in the part 3.1 is interesting. However, it should be proven using additional pharmacological tools, such as TP receptor antagonists or inhibitors of endocannabinoid degradation. Only one example: how can Authors be sure that “constrictor prostanoids contribute to the vascular effects in WT but not in CB1R-KO mice” without the determination of their levels or without using specific antagonists of respective receptors? Especially, Authors did not find any effects of CB1R antagonists on the contraction of mouse aorta induced by PGF2α (Szekeres et al., 2015); this effect was also not mentioned in the current manuscript).

We thank for raising this question, which is really important and must be explained. In the vascular tissue various AA metabolites such as PGs and TXs can act as vasodilators or vasoconstrictors to modulate vascular tone in both physiological and pathophysiological condition (Zhou et al. 2021 Nov 6;22(21):12029, ref 25). PGI2, the principal AA metabolite, is mainly produced by platelets and vascular ECs inducing vasodilatation. Endothelial cells produce COX metabolites including PGD2, PGE2, PGF2α, PGI2 and thromboxanes (TXA2) targeting receptors on vascular smooth muscle among which mainly PGI2, PGD2 and PGE2 are vasodilators, PGF2α and TXA2 are vasoconstrictors acting on FP/TP receptors which was demonstrated in previous studies also by us as mentioned by our Reviewer (Zhou et al. 2021, ref 25, Feletou, M.; et al. The thromboxane/endoperoxide receptor (TP): The common villain. J. Cardiovasc. Pharmacol. 2010, 55, 317–332, Szekeres et al 2015, ref 5). Among these PG metabolites PGI2 and TXA2 are supposed to be the most important in the balance of which determine most of all the vascular homeostasis of PGs (Kij et al. Simultaneous quantification of PGI2 and TXA2 metabolites in plasma and urine in NO-deficient mice by a novel UHPLC/MS/MS method.Journal of Pharmaceutical and Biomedical Analysis Volume 129, 10 September 2016, Pages 148-154).

We thank for our Reviewer for the suggestion to extend the investigation with pharmacological tools to demonstrate directly the presence / effects of constrictor prostanoids to support our conclusion. We previously applied a selective TXA2/PGH2 blocker SQ29548 in coronary arteries of rats in comparison to the nonselective COX blocker indomethacin. We have found that indomethacin decreased vascular tone in coronary arteries of rats and SQ 29548 gave a similar decrease in vascular tone with a less extent. In this study we concluded that constrictor prostaglandins could be identified at least in part as PGH2/TXA2, since SQ29548 elicited increases in diameter, albeit less than indomethacin, suggesting the contribution of other constrictor metabolites of arachidonic acid (Szekeres et al. 2004, ref 24.).

Concerning our previous observation (Szekeres et al. 2015) mentioned by our Reviewer, we agree that we could not modify PGF2 alpha-induced vasoconstriction in mouse aorta with a CB1R blocker. The explanation of this is that PGF2 alpha inducing calcium-independent Rho-dependent vasoconstriction, the mechanism of calcium-mobilizing hormone-induced endocannabinoid release via the activation of diacylglycerol lipase soes not apply in this case. This fact explains the result that CB1R blocker did not modify this contractile effect. On the other hand, in the present study a different pharmacological protocol was examined: role of NO and PGS in the Ach-induced relaxation. Basically, Ach on endothelial cells via muscarinic receptors induce calcium signal and produce the release of NO primarily. However, we were also interested in the role of prostanoid system in the relaxation response, thus we applied nonselective inhibitor of COX, indomethacin. Since it inhibits several relaxant and constrictor PGs, from the change of the tone we may conclude the dominancy of them, not the specific PG types, which mechanisms would be an interesting issue to address in future studies. This, we have corrected the statements when “constrictor prostanoids” are mentioned from these results to “the dominance of constrictors prostanoids is supposed”.

Also, based on the present study in our immunohistochemical results we found a lower COX2 density in the CB1R KO group, but an increased eNOS density compared to WT mice. Based on all these results, and the production of both constrictor (TXA2, TXB2, PGF2a) and relaxant prostanoids (PGE2, PGI2) by COX-2, we believe that the proportion of constrictor prostanoids in the "prostanoid cocktail" is relatively reduced in the CB1R-deficient state. This view is supported by the fact that TP receptor density is not altered in our study between groups.

We have imported some of these facts into the manuscript (Discussion section 3.2. line 431-443)

  1. Part 3.3 is too general, especially in light of very modest effects observed in the current study (e.g., relaxation by WIN55212-2 by about 12%), the examination of the conduit aorta but not resistance artery (e.g., mesenteric), and the fact that vascular CB1R does not play a central role in basal vascular health maintenance (e.g., Wang et al., 2022). This part seems like a very short review.

We agree with our reviewer that CB1Rs do not play a central role in the vascular control mechanisms, though CB1Rs and endocannabinoid system may become more important/pronounced in CV pathologies such as hypertension or ischemia (Wang et al. 2022, ref 60, Batkai et al. 2004, ref 11). We also have found that in vivo during Ang II infusion-induced blood pressure elevation the effect of CB1R antagonist (O2050) is observable on blood pressure (in a further rise) which could not be observed in CB1R ko mice (Szekeres et al. 2015, ref 5). Related to the vasodilatory effects via CB1Rs it is observed in several vascular beds such as in aorta and resistant vessels like in coronary and mesenteric vessels during experimental conditions. Our results on aorta (approx. 10 % relaxation effect), which also corresponds previous studies on aorta (e.g. Dannert et al 2007, Szekeres et al. 2015 , refs 5,46), still, on resistance vessels a higher relaxation can be observed such as in mesenteric arteries (White and Hiley 1998 ref 51) and in gracilis arterioles (Szekeres et al. 2012 ref 14).

We discuss this fact in the Discussion as suggested by our Reviewer (Discussion, lines 387-391).

We have to note that though we have included new information in the Discussion suggested by our Reviewer, we had to make a shortening of the Discussion by the request of our other Reviewer.

Minor comments: 

  1. The solvents for all compounds used in functional studies should be given in chapter 4.1. Moreover, the vascular effects of solvents should be examined. For example, WIN55212-2 requires other solvent than saline solution or water. It caused small relaxation by about 12%. The effect of the solvent is very important in this case.

We have specified all solvents in paragraph 4.1, the methods for dilution and the vehicle constructed by the dilution of DMSO (lines 577-583 in 4.1.) Win 55212 was also compared with vehicle in this latter no relaxation was obtained (we specify this fact in Section 4.1). The amount of Win 55-induced relaxation is explained above in answer No. 5. 

  1. Do antagonists/inhibitors affect basal tension by themselves? The detailed values of basal tone in the presence of antagonists/inhibitors should be given in the manuscript. For example, the NOS inhibitor often increases basal tone.

We have calculated inhibitor responses on the baseline tone. We have found that “vehicle”, INDO and LNA effects did not change the baseline contraction force significantly as calculated by “inhibitor effect – baseline in mN force). We have indicated these values in the Results (lines 175-177).

  1. Statistical analysis – please verify in the Text and in the Legends, and justify the use of two-way ANOVA with Bonferroni post hoc test. It was also listed in the case of comparison of two groups only, e.g., in Fig. 1.

We used 2-way ANOVA with the Bonferroni test to compare groups and the effect of the presence of the inhibitor at different agonist concentrations. We indicate this information in the text (Methods 4.6 and in legends 1-3).

  1. All legends are too long. Authors should try to describe the whole Figures but not particular panels in the frame of the same figure. However, the meaning % in the case of contraction and relaxation should be explained in Legends.

We have shortened the legends as possible even to have important remaining information on them. We have added the information about contraction-relaxation % calculation into the Legends. Our policy is that the Figures should be understandable even without turning to the text.

Reviewer 2 Report

Comments and Suggestions for Authors

The manuscript by Bányai et al. entitled “Role of CB1 cannabinoid receptors in vascular responses and 2 vascular remodeling of the aorta in female mice” is focused on a hot topic of CB1R involvement in the regulation of vascular functions and its interplay with other mediators, including estrogen system, regarding this process. Considering the fact that a large number of CB1R antagonists aimed at the treatment of metabolic disorders, their complications, and CVDs are currently at different stages of preclinical and clinical development, understanding their mechanisms of action and possible interactions with other body regulatory systems are of high importance for both the choice of indication and assessment of safety risks. The manuscript provides very relevant results on sex-related features of CB1R involvement in the regulation of vascular responses and vascular remodeling of the aorta.

I have no remarks regarding the design of the study, the obtained results, and their interpretation, as they are pretty solid and well-described. However, I have some minor suggestions:

1) The authors should correct the title of Figure 3: “Comparison of the effects …. in WT (CB1R +/+) and CB1R KO (CB1R -/-) female mice”. This title misleads as the effects were observed not in mice but ex vivo in the aortic segments of these mice (as described in the case of Figure 1, for example).

2) The discussion section is too large and contains some information already represented in the Introduction. In particular, this applies to the first few paragraphs of section 3.2, where the discussion of the findings of the current study is absent at all. This information can be presented more concisely to make the overall section easier to perceive.

3) The authors should be more careful with their conclusions. They declare from their results that the suppression of the endocannabinoid system by the inhibition of CB1Rs may exert beneficial vascular effects by improving vasodilation and promoting favorable vascular remodeling. But generally speaking, the use of CB1R KO animals is not the same as the inhibition of CB1Rs, since the restructuring of the described processes could occur from the very beginning of the life cycle, including embryonic development. The authors should rephrase this statement and maybe propose that it would be a great aim for future research to study whether the chronic inhibition of CB1R leads to the same effects, and that their current findings reveal the main processes that are worth paying attention to.

Concluding, I recommend the manuscript for publication after the minor corrections.

Comments on the Quality of English Language

Despite the fact that the manuscript in general is written in good and clear language, it still has some typos that should be corrected. Minor editing of English language is required.

Author Response

Answers for Reviewer 2:

We thank for our Reviewer for the useful criticism and suggestions, which we believe have significantly improved the quality of our manuscript. We have answered point by point according to the suggestions and we have implemented them in the text.

The manuscript by Bányai et al. entitled “Role of CB1 cannabinoid receptors in vascular responses and 2 vascular remodeling of the aorta in female mice” is focused on a hot topic of CB1R involvement in the regulation of vascular functions and its interplay with other mediators, including estrogen system, regarding this process. Considering the fact that a large number of CB1R antagonists aimed at the treatment of metabolic disorders, their complications, and CVDs are currently at different stages of preclinical and clinical development, understanding their mechanisms of action and possible interactions with other body regulatory systems are of high importance for both the choice of indication and assessment of safety risks. The manuscript provides very relevant results on sex-related features of CB1R involvement in the regulation of vascular responses and vascular remodeling of the aorta.

I have no remarks regarding the design of the study, the obtained results, and their interpretation, as they are pretty solid and well-described. However, I have some minor suggestions:

1) The authors should correct the title of Figure 3: “Comparison of the effects …. in WT (CB1R +/+) and CB1R KO (CB1R -/-) female mice”. This title misleads as the effects were observed not in mice but ex vivo in the aortic segments of these mice (as described in the case of Figure 1, for example).

We have corrected the title of Figure 3 as suggested to make it more explicit.

2) The discussion section is too large and contains some information already represented in the Introduction. In particular, this applies to the first few paragraphs of section 3.2, where the discussion of the findings of the current study is absent at all. This information can be presented more concisely to make the overall section easier to perceive.

We have made specific corrections in Discussion 3.2 section as suggested by our Reviewer.

We have reduced Introduction and transferred some of these facts into Discussion section 3.2 to avoid duplicates as possible. We also extended Discussion section 3.2 to discuss current findings and we have made reductions in the sections of Discussion to present the information more concisely and also to provide the information suggested by our Reviewers.

3) The authors should be more careful with their conclusions. They declare from their results that the suppression of the endocannabinoid system by the inhibition of CB1Rs may exert beneficial vascular effects by improving vasodilation and promoting favorable vascular remodeling. But generally speaking, the use of CB1R KO animals is not the same as the inhibition of CB1Rs, since the restructuring of the described processes could occur from the very beginning of the life cycle, including embryonic development. The authors should rephrase this statement and maybe propose that it would be a great aim for future research to study whether the chronic inhibition of CB1R leads to the same effects, and that their current findings reveal the main processes that are worth paying attention to.

Our reviewer is right, lacking receptors from fetal life onward is not identical with temporal, even chronic inhibition of them. However, we have a good reason to think that many effects of shutting  off the function of the receptor with chronically administered antagonists will be similar to those observed in our KO animals.  An example of this position is Wang et al 2022 (ref 60) using a conditioned receptor-deficient animal model, also on cardiovascular topics. However, accepting the criticism of our reviewer in the revised version we moderated our statement as required (Conclusion, line 710).

Round 2

Reviewer 1 Report

Comments and Suggestions for Authors

Bányai and co-workers decided to determine the role of the endocannabinoid system in vascular responses and vascular remodeling of the aorta in female mice. Despite Authors revised the Manuscript, in my opinion the study is still affected by several shortcomings, wrong interpretations and some issues need careful and deeper clarification. Just to mention few.

1.   There are two opposite statements in the Abstract:

-      “Cannabinoid type1 receptors (CB1R) mediate vasodilator and hypotensive effects (line 22)” and

-      “Our results indicate that absence or inhibition of CB1Rs may have beneficial vascular effects (line 36).”

2.   Only four lines from the whole 17 lines of the Abstract are devoted to Results.

3.   The aim of the study was determine the impact of endocannabinoid system and CB1 receptor activation on vascular functions, while in my opinion Authors examined only CB1 receptor-dependent effects using knockout CB1-/- animals. 4.   In the functional study there is lack of the analysis of any pharmacological parameters as pEC50, Emax and over-interpretation, i.e. 12% vasorelaxation mediated by WIN55212-2  stayed as “significant” (according to the unpaired t-test it is unsignificant difference; p=0.0700; Fig. 1E). Why did Authors examine only one concentration of WIN55212-2?   5.   The results interpretation, discussion and final conclusions are still too speculative in the context of the involvement of “constrictor prostanoids” in the relaxation responses. Only two examples: (1) In the previous publication Szekeres et al., 2015 demonstrated that WIN-55212,2 relaxed aorta of wild type male mice but not of CB1-/- knockout male mice. The similar results regarding WIN-55212,2 are suggested by Authors in aorta of female mice (the current manuscript). However, quite opposite results have been obtained regarding responses to vasoconstrictors  (enhancement of vasoconstrictor responses to phenylephrine and Angiotensin II but not prostaglandin F2alpha in Szekeres et al., (2015) and no changes to responses induced by phenylephrine and Angiotensin II in the current manuscript) and vasodilator acetylcholine (Szekeres et al. (2015): “Inhibition of CB1R with O2050 did not interfere with the vasodilator effect of acetylcholine (endothelium dependent vasodilator, Supplementary Fig. S5)” the current manuscript – enhancement in CB1-/-). (2) Indomethacin strongly affected the vasodilatory effect of Ach at concentrations of 0.1 and 1 uM whereas in CB1-/- mice the weak enhancement of the vasodilatory effects of Ach was noticed for its 0.01 and 0.1 uM. How can Authors to explain this difference writing about the involvement of “constrictor prostanoids”. Authors wrote in their response that “we believe that the proportion of constrictor prostanoids in the "prostanoid cocktail" is relatively reduced in the CB1R-deficient state.” I can only repeat my first opinion that additional experiments are need to explain this point. Unfortunately, Authors answered for my request describing their previous experiments performed on other vessels of male rats or mice but not writing about additional experiments. Only one example: “In our previous studies we have observed that constrictor prostanoids were released together with the vasodilatory NO, their effect was reduced in trained animals [24,56]. (lines 410-411); however, the above experiments with the use of TP receptor antagonist were performed on small intramural coronary arterioles of male Wistar rats. Moreover, we can still read in the current version of the manuscript
” the observed presence of constrictor prostanoids in the aorta of wild type mice
has been missing in CB1R KO animals” (Lines 406-407) – unfortunately it is still speculation only.

6.   In the new chapter of the Discussion 3.6  “Gender differences in vascular responses of estrogens and cannabinoids” one can read about female animals only. Why authors wrote about gender differences while they did not examine both sexes and did not compare obtained results in females with males mice.

7.   Throughout the text there is the inconsistency in the units of M, M/lit, mol/l (figures), µM, etc.

8.   There are many linguistic errors and misinterpretations, Authors use CB1-/- knockout mice only and wrote i.e. in the final conclusion “  suppression of the endocannabinoid system by the inhibition of CB1Rs may exert beneficial vascular effects by improving vasodilation and promoting favourable vascular remodeling.

Author Response

Bányai and co-workers decided to determine the role of the endocannabinoid system in vascular responses and vascular remodeling of the aorta in female mice. Despite Authors revised the Manuscript, in my opinion the study is still affected by several shortcomings, wrong interpretations and some issues need careful and deeper clarification. Just to mention few.

We thank for our reviewer for the huge effort to improve our manuscript and we have tried to answer each statement. As requested, we also have made extra experiments regarding constrictor PGs to support the statement. We have performed myography in which we have repeated Ach-induced concentration responses with specific TP receptor inhibitor SQ 29,548 (Suppl Fig 1), which gave us similar results as indicated with INDO (in Fig 2) which confirms our Conclusion on augmented role of constrictor PGs in aortas of WT animals compared to KO. We also have performed thromboxane synthase immunostaining which indicates an increasing tendency in its role in WT compared to KO animals (Suppl Fig 2).

We hope that our revised manuscript with extended data will be eligible for the standards of the journal.

  1. There are two opposite statements in the Abstract:

-      “Cannabinoid type1 receptors (CB1R) mediate vasodilator and hypotensive effects (line 22)” and

-      “Our results indicate that absence or inhibition of CB1Rs may have beneficial vascular effects (line 36).”

We have rephrased this issue in the abstract to omit the possible contradiction (abstract lines 28-29).

  1. Only four lines from the whole 17 lines of the Abstract are devoted to Results.

Thank you for the note. We have extended the interpretation of the results keeping the word limit (Lines 40-42) .

  1. The aim of the study was determine the impact of endocannabinoid system and CB1 receptor activation on vascular functions, while in my opinion Authors examined onlyCB1 receptor-dependent effects using knockout CB1-/- animals. 

Genetically defective animals are the proper tools to reveal the functional significance of a gene product throughout the life. Our reviewer is right. This situation is not identical with (chronic) inhibition of the function in a certain phase of life.  However, we have a good reason to think that  observations on genetically knocked out animals will be, to some degree, valid for the situation with pharmacological inhibition.  We refrased the Aims to separate the two situations.

  1. In the functional study there is lack of the analysis of any pharmacological parameters as pEC50, Emaxand over-interpretation, i.e. 12% vasorelaxation mediated by WIN55212-2  stayed as “significant” (according to the unpaired t-test it is unsignificant difference; p=0.0700; Fig. 1E). Why did Authors examine only one concentration of WIN55212-2?   

We thank for the suggestion. We have provided also pharmacological parameters pEC50, Emax and pEC50 for the inhibitor responses (Fig 2) in Supplementary Table 1.  

WIN55212 vasodilation was significant in our study, p=0,033 (Results, line 161). We have used WIN in one effective concentration (10-5 mol/l) which was experienced previously, because this drug needs a longer equilibration to achieve stable pharmacological response.

  1. The results interpretation, discussion and final conclusions are still too speculative in the context of the involvement of “constrictor prostanoids” in the relaxation responses. Only two examples: (1) In the previous publication Szekeres et al., 2015 demonstrated that WIN-55212,2 relaxed aorta of wild type male mice but not of CB1-/-knockout male mice. The similar results regarding WIN-55212,2 are suggested by Authors in aorta of female mice (the current manuscript). However, quite opposite results have been obtained regarding responses to vasoconstrictors  (enhancement of vasoconstrictor responses to phenylephrine and Angiotensin II but not prostaglandin F2alpha in Szekeres et al., (2015) and no changes to responses induced by phenylephrine and Angiotensin II in the current manuscript) and vasodilator acetylcholine (Szekeres et al. (2015): “Inhibition of CB1R with O2050 did not interfere with the vasodilator effect of acetylcholine (endothelium dependent vasodilator, Supplementary Fig. S5)” the current manuscript – enhancement in CB1-/-). (2) Indomethacin strongly affected the vasodilatory effect of Ach at concentrations of 0.1 and 1 uM whereas in CB1-/- mice the weak enhancement of the vasodilatory effects of Ach was noticed for its 0.01 and 0.1 uM. How can Authors to explain this difference writing about the involvement of “constrictor prostanoids”.

The above mentioned previous studies were all performed on male animals where we have found that CB1R inhibition augmented vasoconstrictor responses induced by Gq signaling-dependent DAG lipase mediated endocannabinoid release which have attenuated the constrictor response. In the present study in female mice the situation is different. We have used only genetic methods (CB1R ko mice) in which the constrictor responses were similar, but still, CB1R-induced vascular responses could be detected by WIN55,212. Since several other mechanisms are involved in the vascular control, we suggested that the proportion of NO and PGs may also have changed in females compared to males. This issue we try to discuss in section 3.6, Discussion. We have added a new reference to refer to contractile differences in genders (Varbiro et al. 2006, ref 79)

Authors wrote in their response that “we believe that the proportion of constrictor prostanoids in the "prostanoid cocktail" is relatively reduced in the CB1R-deficient state.” I can only repeat my first opinion that additional experiments are need to explain this point. Unfortunately, Authors answered for my request describing their previous experiments performed on other vessels of male rats or mice but not writing about additional experiments. Only one example: “In our previous studies we have observed that constrictor prostanoids were released together with the vasodilatory NO, their effect was reduced in trained animals [24,56]. (lines 410-411); however, the above experiments with the use of TP receptor antagonist were performed on small intramural coronary arterioles of male Wistar rats. Moreover, we can still read in the current version of the manuscript
” the observed presence of constrictor prostanoids in the aorta of wild type mice
has been missing in CB1R KO animals” (Lines 406-407) – unfortunately it is still speculation only.

For the suggestion of our Reviewer we have repeated vascular experiments with a specific TP receptor blocker (SQ 29,548) and also we have performed immunohistochemistry to detect thromboxane synthase levels with specific antibody, which extra data show directly the involvement of most of the constrictor PGs acting on TP receptors (TXA2/PGH2) as also mentioned above (Szenasi et al. 2021, Suppl reference 1). We have added these new data in the Supplement to support our Conclusion.

We hope that with the addition of these new results it is more relevant to mention the role of „constrictor PGs” in our study.

  1. In the new chapter of the Discussion 3.6  “Gender differences in vascular responses of estrogens and cannabinoids” one can read about female animals only. Why authors wrote about gender differences while they did not examine both sexes and did not compare obtained results in females with males mice.

As we mentioned also above, we have added a new reference to refer to contractile differences in genders (Varbiro et al. 2006, ref 79)

  1. Throughout the text there is the inconsistency in the units of M, M/lit, mol/l (figures), µM, etc.

We have corrected these issues in the text.

  1. There are many linguistic errors and misinterpretations, Authors use CB1-/-knockout mice only and wrote i.e. in the final conclusion “  suppression of the endocannabinoid system by the inhibition of CB1Rs may exert beneficial vascular effects by improving vasodilation and promoting favourable vascular remodeling.

We have corrected this interpretation error. Explanation please see at point 3.

Round 3

Reviewer 1 Report

Comments and Suggestions for Authors

The authors have provided satisfactory replies to my comments and I have no further comments.